# Patterns of Heterochromatin Transitions Linked to Changes in the Expression of *Plasmodium falciparum* Clonally Variant Genes

Lucas Michel-Todó,[a] Cristina Bancells,[a] Núria Casas-Vila,[a] Núria Rovira-Graells,[a] Carles Hernández-Ferrer,[b,c] Juan R. González,[b,c] Alfred Cortés[a,d]

aISGlobal, Hospital Clínic, Universitat de Barcelona, Barcelona, Catalonia, Spain
bISGlobal, Barcelona, Catalonia, Spain
cCentro de Investigación Biomédica en Red en Epidemiología y Salud Pública (CIBERESP), Madrid, Spain
dICREA, Barcelona, Catalonia, Spain

**ABSTRACT** The survival of malaria parasites in the changing human blood environment largely depends on their ability to alter gene expression by epigenetic mechanisms. The active state of *Plasmodium falciparum* clonally variant genes (CVGs) is associated with euchromatin characterized by the histone mark H3K9ac, whereas the silenced state is characterized by H3K9me3-based heterochromatin. Expression switches are linked to euchromatin-heterochromatin transitions, but these transitions have not been characterized for the majority of CVGs. To define the heterochromatin distribution patterns associated with the alternative transcriptional states of CVGs, we compared H3K9me3 occupancy at a genome-wide level among several parasite subclones of the same genetic background that differed in the transcriptional state of many CVGs. We found that *de novo* heterochromatin formation or the complete disruption of a heterochromatin domain is a relatively rare event, and for the majority of CVGs, expression switches can be explained by the expansion or retraction of heterochromatin domains. We identified different modalities of heterochromatin changes linked to transcriptional differences, but despite this complexity, heterochromatin distribution patterns generally enable the prediction of the transcriptional state of specific CVGs. We also found that in some subclones, several *var* genes were simultaneously in an active state. Furthermore, the heterochromatin levels in the putative regulatory region of the *gdv1* antisense noncoding RNA, a regulator of sexual commitment, varied between parasite lines with different sexual conversion rates.

**IMPORTANCE** The malaria parasite *P. falciparum* is responsible for more than half a million deaths every year. *P. falciparum* clonally variant genes (CVGs) mediate fundamental host-parasite interactions and play a key role in parasite adaptation to fluctuations in the conditions of the human host. The expression of CVGs is regulated at the epigenetic level by changes in the distribution of a type of chromatin called heterochromatin. Here, we describe at a genome-wide level the changes in the heterochromatin distribution associated with the different transcriptional states of CVGs. Our results also reveal a likely role for heterochromatin at a particular locus in determining the parasite investment in transmission to mosquitoes. Additionally, this data set will enable the prediction of the transcriptional state of CVGs from epigenomic data, which is important for the study of parasite adaptation to the conditions of the host in natural malaria infections.

**KEYWORDS** ChIP-seq, H3K9me3, *Plasmodium falciparum*, chromatin, *gdv1*, clonally variant gene expression, epigenetics, heterochromatin, malaria, transcriptomics

Address correspondence to Alfred Cortés, alfred.cortes@isglobal.org.

The authors declare no conflict of interest.

The apicomplexan parasite *Plasmodium falciparum*, the causative agent of malaria, has a complex life cycle divided between female *Anopheles* mosquitoes and humans. Human infection starts with the injection of parasites at the sporozoite stage during an

*Anopheles* blood meal. After multiplication in the liver, thousands of parasites at the merozoite stage are released into the bloodstream. Merozoites invade red blood cells (RBCs) and start continuous rounds of the asexual intraerythrocytic development cycle (IDC), in which parasites develop through the intraerythrocytic ring, trophozoite, and schizont stages and then burst and release up to 32 new merozoites. Repeated rounds of the ~48-h IDC are responsible for all of the clinical manifestations of malaria. At each round, some parasites differentiate into sexual forms called gametocytes, which, after developing for ~10 days, can infect an *Anopheles* mosquito during a new blood meal. In the mosquito vector, male and female gametocytes mate and form zygotes that, after meiosis and several asexual replication steps, generate new sporozoites (1).

The complex life cycle of *P. falciparum* requires adaptation to the different conditions of each niche, which is in large part regulated at the transcriptional level. Therefore, the parasite has a different transcriptome at each stage of development (2–4). In addition, the parasite must adapt to fluctuations in the conditions of each niche. In the human blood, where infections can last for several months, parasites are exposed to changes in the availability of nutrients, in temperature (associated with fever episodes), in the presence of toxic compounds such as antimalarial drugs, and in host immune responses, among others (5–8). Rapid adaptation to fluctuations in the conditions of the human blood can be achieved by directed transcriptional responses, which involve sensing followed by changes in gene expression that protect against the challenge. However, malaria parasites can produce directed transcriptional responses to only a limited number of conditions (9–12). An alternative adaptive mechanism to survive under fluctuating conditions is based on epigenetic variation. In *P. falciparum*, epigenetic variation is linked to clonally variant genes (CVGs), which can be found in different transcriptional states (active or silenced) in parasites living in the same environment, with the same genome sequence, and at the same stage of the IDC. The transcriptional state of CVGs is transmitted from one round of the IDC to the next, but low-frequency switches between the active and silenced states occur, ensuring the constant generation of epigenetic diversity within parasite populations (13–17). As a consequence of this expression dynamics, different individual parasites within an isogenic population have different combinations of active and silenced CVGs. Transcriptional heterogeneity in parasite populations results in phenotypic heterogeneity, which provides the grounds for the continuous natural selection of epigenetic variants best adapted to the dynamic conditions of the environment. This constitutes a bet-hedging adaptive strategy (18).

Essentially all CVGs are involved in host-parasite interactions (13–17). The best-characterized family of CVGs is the *var* gene family, consisting of ~60 genes that encode *P. falciparum* erythrocyte membrane protein 1 (PfEMP1), a major virulence factor. PfEMP1 is exposed at the surface of infected RBCs and mediates antigenic variation and adhesion to endothelial cell receptors, which results in parasite sequestration within different host tissues. Individual parasites generally express a single member of the *var* gene family in a mutually exclusive manner, which is important for immune evasion (19–21). Other gene families linked to processes such as antigenic variation (e.g., *rifin* and *stevor*) (21), RBC remodeling (e.g., *pfmc-2tm* and *phist*) (22, 23), RBC invasion (e.g., *eba* and *Rh*) (24), lipid metabolism (e.g., *acs* and *acbp*) (25), or solute transport and nutrient acquisition (*clag*, with *clag3.1* and *clag3.2* displaying mutually exclusive expression) (26–29), among others, also show clonally variant expression (13, 18, 19, 26, 30–33). A peculiar CVG is the *pfap2-g* gene, which encodes a transcription factor of the ApiAP2 family that drives the conversion of asexual forms into nonreplicative sexual gametocytes (34–36). This gene is silenced in asexually growing parasites and, upon activation, triggers a transcriptional cascade that results in sexual conversion (37, 38). The proportion of parasites that activate *pfap2-g* at each round of the IDC determines the proportion of parasites that convert into sexual forms, i.e., the sexual conversion rate.

The epigenetic regulation of the expression of *P. falciparum* CVGs is based on reversible chromatin transitions between euchromatin (active) and facultative heterochromatin (silenced) states (13–17). The heterochromatic state is associated with the

histone posttranslational modification histone H3 trimethylation of lysine 9 (H3K9me3) and the associated heterochromatin protein 1 (PfHP1) (39–42). Furthermore, several ApiAP2 DNA binding proteins such as PfAP2-HC are also associated with heterochromatic regions (43, 44). The active state of CVGs is characterized by the absence of H3K9me3 and PfHP1 as well as the acetylation of H3K9 (H3K9ac) (45–48). The chromatin at CVG loci is considered bistable because, once established, both the euchromatic and the heterochromatic states are maintained across the full IDC and can be transmitted to the next generations. Infrequent transitions between the euchromatic and heterochromatic states at these loci underlie the expression switches of CVGs (15, 16).

In higher eukaryotes, H3K9me3 is typically associated with constitutive heterochromatin localized in subtelomeric and pericentromeric repetitive regions, but recent research has demonstrated that it also participates in cell differentiation processes (49, 50). In *P. falciparum*, H3K9me3 and PfHP1 are absent from pericentromeric regions, but they occupy subtelomeric repeats in addition to CVG loci. CVGs are located mainly in subtelomeric regions and a few chromosome internal clusters, and there are also a few stand-alone CVG loci in internal chromosome regions (40, 42, 44, 51, 52). In total, over 500 genes (~10% of the genome) carry heterochromatin marks and are considered CVGs. In contrast to the restricted distribution of H3K9me3, the histone mark associated with the active state of CVGs, H3K9ac, is widespread across the *P. falciparum* genome and marks constitutively euchromatic genes in addition to active CVGs. In non-CVGs, H3K9ac levels correlate with transcript levels and the temporal patterns of gene expression across the IDC (53, 54).

Previous studies showed that the overall distribution of heterochromatin at CVGs is stable across different stages of the *P. falciparum* IDC (40, 45–47, 52) but changes during transmission stages (52, 55–57). Indeed, in contrast to the relatively stable expression patterns of CVGs during multiple rounds of the IDC, the expression patterns of CVGs are reset during transmission stages (58–60). The overall distribution of heterochromatin is similar across *Plasmodium* spp., suggesting that it plays similar roles across the genus (52). A recent study showed that different *P. falciparum* lines have a similar overall distribution of heterochromatin, with differences at specific loci possibly reflecting different transcriptional states of the genes at these loci among the parasite lines analyzed, although gene expression was not assessed (52). However, the parasite lines used in that study had not been recently subcloned. Consequently, they were likely transcriptionally heterogeneous such that for many CVGs, the population is a mixture of parasites that have the gene in an active state and parasites that have it in a silenced state, as previously demonstrated for other parasite lines under long-term culture (18).

Here, we combined heterochromatin profiling with transcriptomic analysis to investigate in detail the chromatin differences between the active and silenced states of CVGs at a genome-wide level. For this, we compared the distribution of H3K9me3 between subcloned lines of the same genetic background that differ in the expression of dozens of CVGs. The use of subclones ensures that most genes are in a homogeneously active or silenced state within each population. We also compared the genome-wide distribution of heterochromatin between parasite lines that produce different numbers of gametocytes, with the aim of identifying epigenetic differences linked to sexual conversion rates.

## RESULTS

**Overview of the study.** We analyzed the genome-wide distribution of H3K9me3 using chromatin immunoprecipitation followed by sequencing (ChIP-seq) and the transcriptome across the full IDC using two-channel microarrays for five different subclones of the 3D7 genetic background: 1.2B, 10G, A7, E5, and B11 (Fig. 1). The transcriptome analysis of the 1.2B and 10G subclones was previously reported (18). The analysis of 5 subclones enables 10 possible pairwise comparisons to identify genes that are active in one subclone and silenced in another. The selection of the subclones was based on

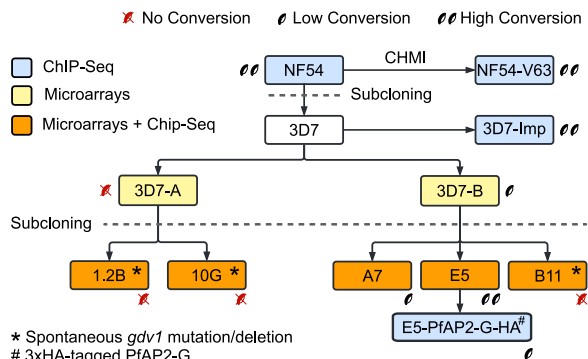

**FIG 1** Schematic of the parasite lines used in this study. 3D7-A and 3D7-B are stocks of the clonal 3D7 line that were maintained in different laboratories for some years. Blue indicates parasite lines that were analyzed by ChIP-seq, yellow indicates parasite lines for which new or previously published transcriptomic data (using microarrays) were generated, and orange indicates parasite lines for which both ChIP-seq and transcriptomic data were generated. Parasite lines with a spontaneous deletion or a premature stop codon in *gdv1* (*) or with a genetic edition in the *pfap2-g* locus (#) are indicated. Parasite lines are semiquantitatively classified as high-gametocyte producers (two gametocytes, >10% sexual conversion rate), low-gametocyte producers (one gametocyte, <3% sexual conversion rate), or non-gametocyte producers (red, crossed gametocytes). NF54-V63 was obtained after human infection with the NF54 line in a controlled human malaria infection (CHMI) trial.

maximizing the number of differentially expressed genes among them and covering a broad range of sexual conversion rates. Subclones 1.2B and 10G were derived from the 3D7-A stock of 3D7, whereas A7, E5, and B11 were derived from the 3D7-B stock. The 3D7-A and 3D7-B stocks were maintained in different laboratories for several years and have phenotypic differences in growth rates, sexual conversion rates, density-dependent growth inhibition, the use of invasion pathways, and cytoadherence (26, 34, 61–64). The basal sexual conversion rate of the subclones ranges from zero (10G, 1.2B, and B11) to intermediate (<3%) (A7) or high (~15%) (E5) (34). To preserve transcriptional homogeneity, subclones were maintained in culture for the shortest time needed to obtain sufficient RNA or chromatin for the analyses (~4 to 5 weeks between limiting dilution to obtain the subclones and epigenomic or transcriptomic analysis).

Furthermore, we analyzed H3K9me3 occupancy in additional parasite lines with different basal sexual conversion rates: an E5-derived transgenic parasite line in which PfAP2-G was 3× hemagglutinin (3×HA) tagged (E5-PfAP2-G-HA) and the sexual conversion rate was lower (<3%) than that in the parental E5 line (34, 65); a stock of 3D7 obtained from Imperial College (3D7-Imp) with a high basal sexual conversion rate (66) (conversion rate in Albumax medium, 18.3% [range, 16.6 to 20%] [*n* = 2]); and the NF54 isolate from which the 3D7 clone was obtained, also a high-gametocyte producer (66) (conversion rate in Albumax medium, 21.8% [range, 18.8 to 24.8%] [*n* = 2]). In some of the analyses, we also included published ChIP-seq data for the NF54-V63 line that was recovered after infecting a volunteer with the NF54 line in a controlled human malaria infection (CHMI) trial (58). We also analyzed the distribution of H3K9ac in several of the parasite lines.

**Transcriptomic analysis of 3D7 subclones.** To identify genes that were transcriptionally active in one subclone and silenced in another, we analyzed the transcriptomes of tightly synchronized cultures (5-h age window) of the A7, E5, and B11 lines at six time points along the IDC. The analysis pipeline to identify genes with large expression differences between the subclones, similar to the method used in our previous study including 10G, 1.2B, and 3D7-B (18), was based on the statistically estimated parasite age (67), the average fold change (AFC) in transcript levels (between subclones) at four overlapping time intervals of the IDC, and the maximum (among the time intervals) average fold change (mAFC) parameter (18, 58). After applying several filters (see Materials and Methods), we generated for each pair of subclones a high-confidence list of differentially expressed genes with an mAFC of >4 [$\log_2$(mAFC) of >2].

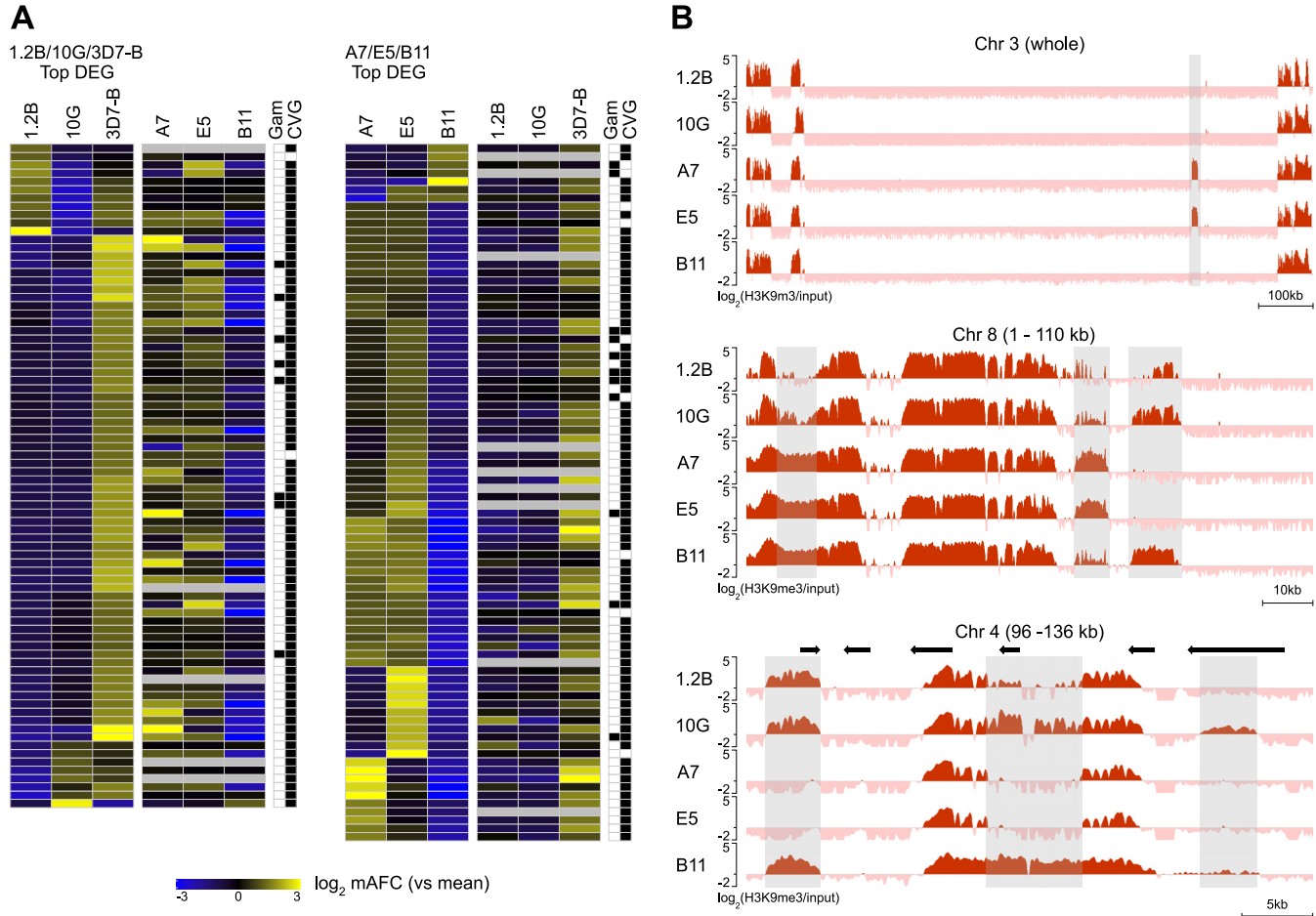

**FIG 2** Overview of the transcriptomic and epigenomic analyses of the subclones. (A) Expression patterns of the top differentially expressed genes (DEG). (Left) Genes with differential expression (mAFC of >4) among the 1.2B, 10G, and 3D7-B lines (previously published data [18]). Values for the expression differences in the same genes among the A7, E5, and B11 lines (newly generated data) are shown. All values are the mean-centered log$_2$ of the mAFC in expression [log$_2$(mAFC)] among the three parasite lines. (Right) Analogous analysis for the top differentially expressed genes in the comparison of A7, E5, and B11. The last two columns indicate whether a gene is a known gametocyte marker or has been previously classified as a CVG. The values presented in this panel are provided in Data Set S2 in the supplemental material. (B) H3K9me3 distribution in a representative chromosome or chromosome regions in the five subclones at different magnification levels. Gray shades indicate regions with major differences in H3K9me3 coverage among subclones. Values are log$_2$(H3K9me3/input).

Between 19 and 74 genes were differentially expressed in the pairwise comparisons between 10G and 1.2B or among A7, E5, and B11 (Fig. 2A; see also Data Sets S1 and S2 in the supplemental material). Since 3D7-A subclones (1.2B and 10G) and 3D7-B subclones (A7, E5, and B11) were analyzed using microarrays with slightly different designs (18, 35, 68) and with different preparations of the reference pool, a direct comparison between subclones of the two different groups was not possible. However, we combined data for A7, E5, and B11 with data for the direct comparison of 1.2B and 10G against 3D7-B (18) to identify genes that were likely differentially expressed between 10G or 1.2B and A7, E5, or B11 (see Materials and Methods). In total, 121 genes were differentially expressed with high confidence in at least one of the possible pairwise comparisons, of which 111 were genes previously classified as CVGs based on the presence of heterochromatin marks or variant expression in previous studies (see the list of CVGs in reference 58). A few of the differentially expressed genes are gametocyte markers, and their different transcript levels may reflect the different sexual conversion rates among the subclones (Fig. 2A).

A large fraction of the differentially expressed genes showed lower expression levels in B11 than in A7 and E5. Unexpectedly, many of these genes were also expressed at lower levels in 1.2B and 10G than in 3D7-B, suggesting that the CVG expression patterns in B11 were relatively similar to those in 1.2B and 10G (Fig. 2A). Analysis of

genetic polymorphisms in these parasite lines using the ChIP-seq input data revealed several large deletions in B11, including an ~20-kb deletion affecting the *gdv1* locus (69, 70) (Fig. S1 and Data Set S3). Of note, 10G and 1.2B also have a defect in *gdv1*, in this case a mutation that results in a premature stop codon (Q578X) (Data Set S3) (35).

**Heterochromatin distribution in the 3D7 subclones and general association with gene expression.** ChIP-seq analysis of H3K9me3 revealed similar global distribution of heterochromatin among the five subclones, although differences were apparent at specific loci (see Fig. 2B for representative examples). The mean pairwise correlation among parasite lines was 0.88 (range, 0.83 to 0.95) using input-normalized coverage values (Fig. S2). We performed individual peak calling using the MACS2 tool (Data Set S4) and differential peak calling for all possible pairwise comparisons using custom scripts. There was an average of 142 (range, 68 to 214) differential peaks in pairwise comparisons, which map to an average of 62 (range, 34 to 99) genes (Data Set S5). Differential peaks involving changes in H3K9me3 coverage at the upstream region (1,000 bp) or the first half of the coding sequence (CDS) of a gene were generally associated with transcriptional differences such that the parasite line with higher coverage had lower transcript levels. However, when H3K9me3 coverage was different in only the second half of the coding sequence or the downstream region of a gene, typically, there were no associated transcriptional differences (see Fig. 3A for the comparison of 10G versus 1.2B and Fig. S3A to C for the other direct pairwise comparisons; Data Set S2). In addition to differences in the heterochromatin occupancy observed at chromosome regions containing genes, there were also differences at the telomere-associated repetitive elements (TAREs) (71, 72) located between the telomeres and subtelomeric genes (Fig. S1).

To further assess the relationship between differences in heterochromatin distribution and transcriptional differences, we first calculated for all differentially expressed genes in any of the pairwise comparisons the level of correlation between the H3K9me3 coverage at different landmark regions of the genes and transcript levels. The H3K9me3 coverage was calculated for fixed-length regions at different positions relative to the start or stop codons and also at the 5′ untranslated region (UTR), the coding sequence, and the 3′ UTR (according to the genome annotation in PlasmoDB v52). The strongest negative correlation between H3K9me3 coverage and transcript levels occurred at the 5′ UTR and in the fixed-length −500 bp to ATG regions, but strong correlations (Pearson's $|r|$ value of >0.6) were observed throughout the upstream and coding sequences (Fig. S4). In genes with large transcriptional differences (mAFC of >4) among the subclones, there was a strong negative association between transcript levels and H3K9me3 coverage at the region from −1000 to +500 bp such that the subclone that expressed a gene at higher levels tended to have less heterochromatin (Fig. 3B, Fig. S3D to F, and Data Set S2).

**Changes in heterochromatin distribution at CVG loci between the active and silenced states.** To characterize in more detail the changes in the heterochromatin distribution that drive the transition between the silenced and active states of CVGs, we analyzed for each gene the H3K9me3 coverage across 23 bins encompassing the neighboring genes, the upstream region (until the first upstream gene), the CDS, and the downstream region (until the first downstream gene). For the 121 genes in the high-confidence list of differentially expressed genes, we compared the heterochromatin distribution between the two subclones with the maximum expression differences (Fig. 4). The subclone with the highest expression level was used to represent the active state of the gene, and the subclone with the lowest expression level was used to represent the silenced state. For genes in which the largest expression difference occurred between the 1.2B or 10G subclone and 3D7-B, with higher expression levels in 3D7-B than in 10G or 1.2B, we used the 3D7-B subclone (A7, E5, or B11) with the highest expression level as the subclone representative of the active state (and vice versa; i.e., for genes with lower expression levels in 3D7-B than in 10G or 1.2B, the 3D7-B subclone with the lowest expression level was used as the representative of the silenced state). The H3K9me3 coverages in the active and silenced states and the coverage difference

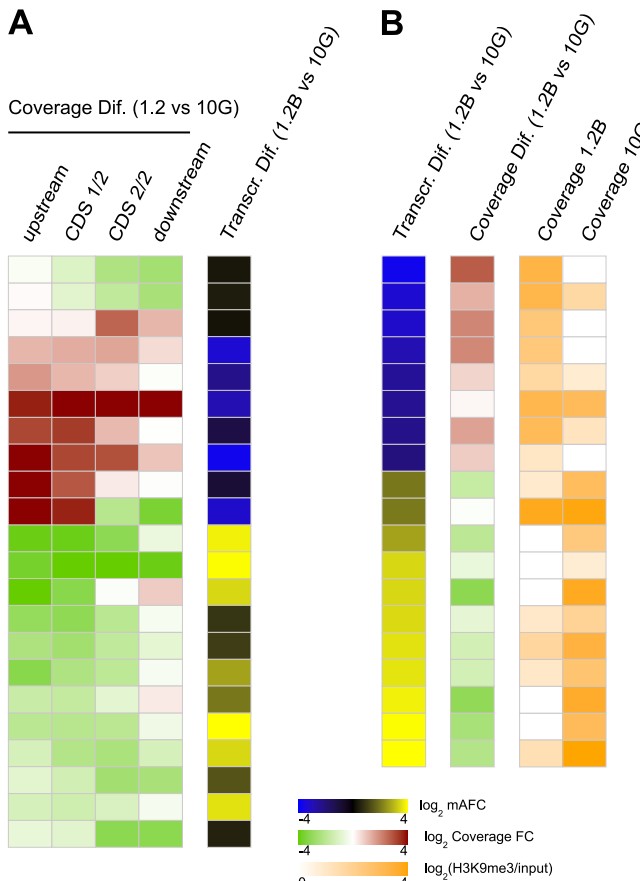

**FIG 3** Association between H3K9me3 coverage differences and transcriptional differences. Results for subclones 1.2B and 10G are shown (see Fig. S3 in the supplemental material for the other pairwise comparisons). (A) Transcriptional differences in genes overlapping (from −1000 to +500 bp relative to the ATG start codon) an H3K9me3 differential peak between 1.2B and 10G. The $\log_2$ of the input-normalized coverage fold change (FC) in 1.2B relative to 10G is shown for the upstream region (1,000 bp before the ATG codon), the first or second half of the coding sequence (CDS 1/2 or CDS 2/2, respectively), and the downstream region (1,000 bp after the stop codon). Transcriptional differences are expressed as the $\log_2$ of the mAFC (1.2B versus 10G). (B) Heterochromatin levels in genes differentially expressed between 1.2B and 10G (mAFC of >4). Transcriptional differences are shown as described above for panel A. Heterochromatin coverage ($\log_2$ of the input-normalized coverage) and coverage differences between 10G and 1.2B ($\log_2$ of the input-normalized coverage fold change) are shown for the region from −1000 to +500 bp relative to the ATG codon. The values presented in this figure are provided in Data Set S2.

between the two were used to group the 121 differentially expressed genes into seven distinct clusters using the *k*-means algorithm (Fig. 4 and Data Set S2).

Cluster 1 (22 genes) corresponds to genes that are largely devoid of heterochromatin in the upstream and coding regions in their active state but fully heterochromatic in these regions and also in the downstream region in their silenced state. We called this the "global transition" cluster (Fig. 4 and Fig. 5A to C). Genes in cluster 2 (14 genes) have heterochromatin patterns that are relatively similar to those of the genes in cluster 1, but in the silenced state, the downstream region remains euchromatic ("mainly 5′ transition" cluster) (Fig. 4 and Fig. 5D and E).

Clusters 3, 4, and 5 (20, 13, and 29 genes, respectively) correspond to genes in which the full locus (clusters 3 and 5) or the upstream region and the beginning of the coding region (cluster 4) are marked by H3K9me3 in the silenced state, and in the active state, there is only a localized reduction of the coverage of this mark in the upstream region (Fig. 4 and Fig. 5F and G). We called these the "localized 5′ transition" clusters. In a few genes from these clusters, changes in gene expression were not accompanied by noticeable differences in heterochromatin levels. We cannot exclude

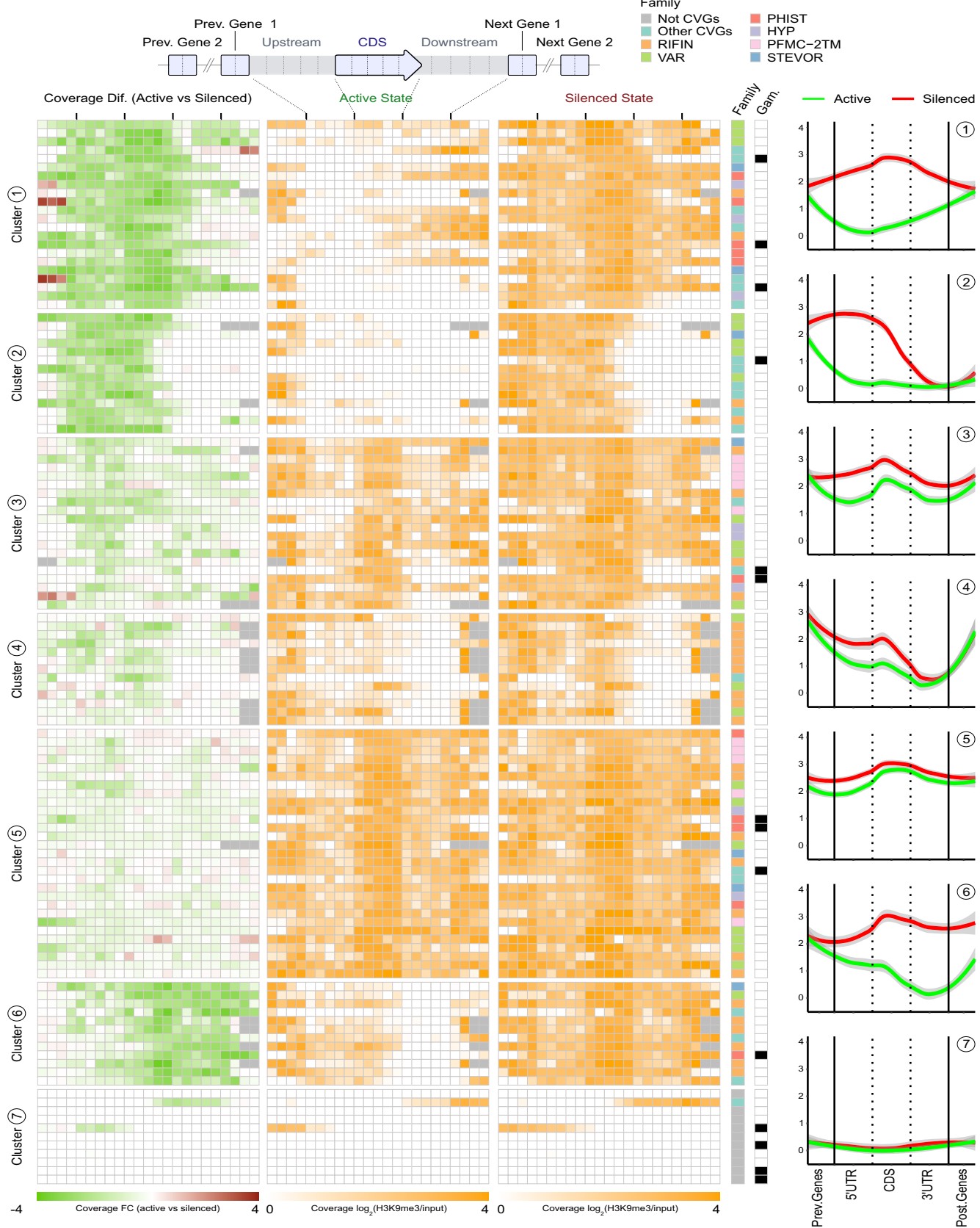

**FIG 4** Distribution of heterochromatin in the active and silenced states of CVGs. The H3K9me3 coverage in the top differentially expressed genes among 1.2B, 10G, A7, E5, and B11 in the active compared to the silenced state is shown. For each gene, the subclone with the highest expression level was

the possibility that, even in recently subcloned lines, some of these genes may be expressed in only a small fraction of the parasites in the subclone in which they are tagged as active.

Cluster 6 (12 genes) includes genes in which heterochromatin is present in the upstream, coding, and downstream regions in the silenced state, whereas in the active state, heterochromatin is depleted mainly in the second half of the coding region and the downstream region ("mainly 3′ transition" cluster) (Fig. 4 and Fig. 5H and I). This pattern suggests that heterochromatin in the downstream region of these genes may play a role in their transcriptional regulation, although we cannot exclude the possibility that their expression was actually driven by differences in the upstream region. Indeed, differences in the upstream region between the active and silenced states occurred in the majority of cluster 6 genes, albeit these were less pronounced than the differences in the downstream region.

Finally, cluster 7 (11 genes) corresponds to genes largely devoid of heterochromatin in both the active and silenced states. Therefore, the transcriptional variation of these genes is not controlled by euchromatin-heterochromatin transitions. Of note, the vast majority of the genes in this cluster had not been previously classified as CVGs.

Except for the non-CVGs from cluster 7, in the majority of the differentially expressed genes, there was heterochromatin at some part of the locus or at the neighboring genes, even in the active state there was heterochromatin at some part of the locus or at neighboring genes. Furthermore, the transition from the active to the silenced state was typically associated with an increase in heterochromatin levels in a continuous region spanning from nearby heterochromatic regions to the regulatory regions of the gene. This is consistent with the silencing and activation of CVGs commonly involving the spreading and retraction of heterochromatin, respectively, according to an accordion-like mechanism. In several cases, this involved changes in the position of the outer limit of the heterochromatin domain (Fig. 5A, D, and H to J), whereas in other cases, the transition from the silenced to the active state was associated with reduced heterochromatin levels in a region surrounded by heterochromatin on both sides. The latter is consistent with an opening at an internal point within the heterochromatin domain followed by heterochromatin retraction (Fig. 5B, F, and G). We defined four main transition patterns between the active and silenced states of CVGs (Fig. 5L) and qualitatively assigned each of the 109 classifiable differentially expressed genes in clusters 1 to 6 to the pattern with which they best matched (Fig. 5L and Data Set S5). Only 5 differentially expressed genes showed transitions involving heterochromatin *de novo* formation or complete removal, whereas 65 genes showed transitions consistent with heterochromatin expansion/retraction, either affecting the external limits of a heterochromatin domain ["expansion/retraction (limits)"] (42 genes) or involving an opening within a heterochromatin domain that was continuous in the silenced state ["expansion/retraction (internal)"] (23 genes). In 39 genes, the transition between the active and silenced states involved small changes affecting heterochromatin levels in only a very small region ("localized closing/opening") (Fig. 5L). However, it is important to note that the four general patterns likely represent a mechanistic continuum rather than discrete, fully independent mechanisms, and for some genes, the classification was ambiguous. Furthermore, in some genes, transitions followed a different pattern depending on the state of the neighboring genes.

**FIG 4** Legend (Continued)

selected as the representative for the active state, and the subclone with the lowest expression level was selected as the representative for the silenced state. The distribution of heterochromatin in each subclone ($\log_2$ of the input-normalized coverage) and the coverage difference ($\log_2$ of the fold change [FC] in coverage in the active versus the silenced state) are shown for 23 bins spanning the coding sequence (CDS) (divided into 5 bins), the upstream and downstream regions (up to the next gene) (5 bins each), and the coding sequences of the two upstream and downstream neighboring genes (2 bins each). Genes were clustered according to their H3K9me3 coverage and coverage differences (excluding the bins from the neighboring genes) using *k*-means clustering (*k* = 7, guided by silhouette-and-elbow plots). Within each cluster, genes are ordered by fold changes in transcript levels between the active and silenced states. The two columns at the right of the heat map indicate the gene family and whether a gene is a known gametocyte marker. The plots on the right are locally estimated scatterplot smoothing (LOESS) regression plots (shades are 95% confidence intervals) showing the heterochromatin distribution ($\log_2$ of the input-normalized H3K9me3 coverage) in the active and silenced states for each cluster. The details of the genes included in this figure are provided in Data Set S2 in the supplemental material.

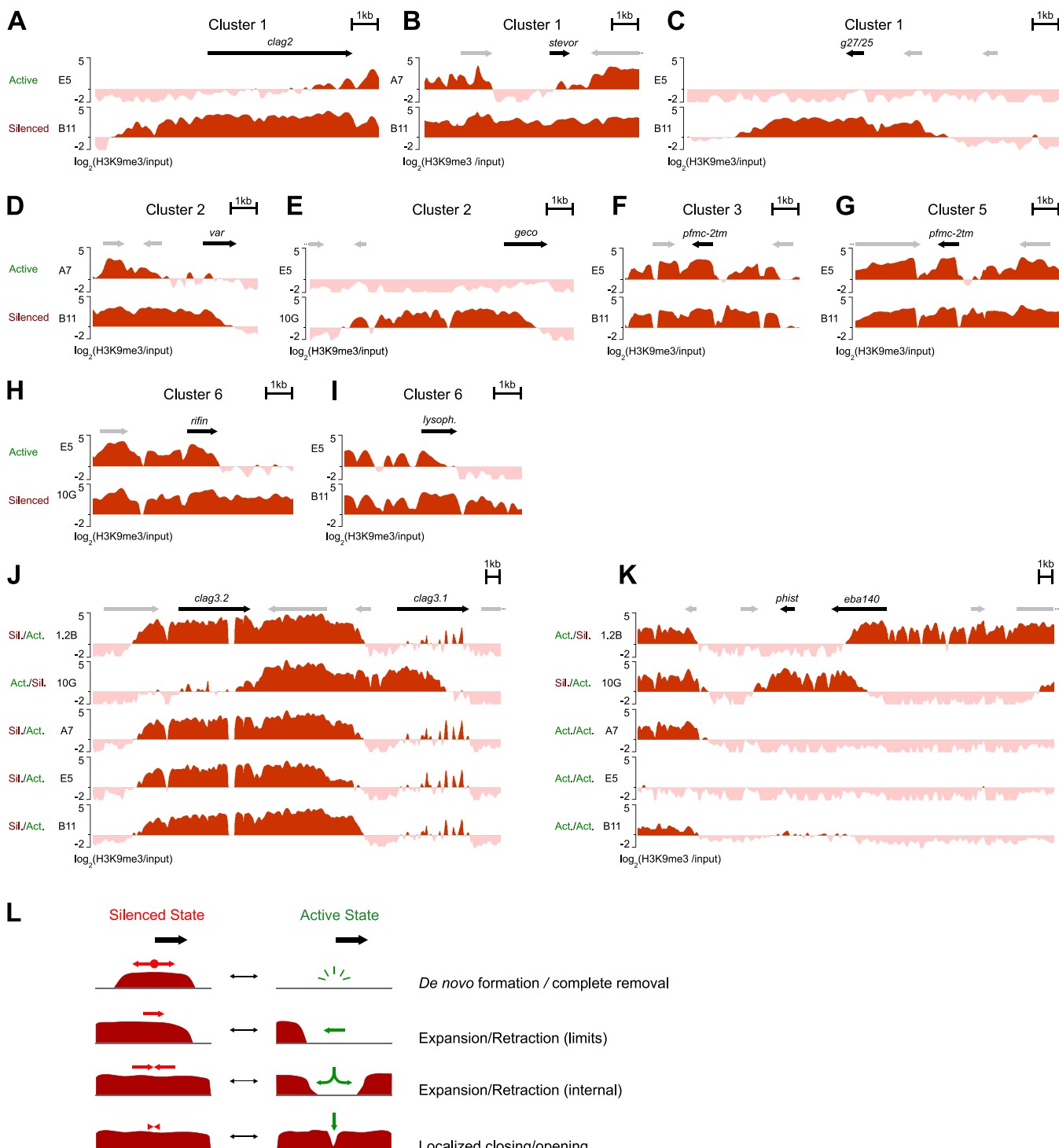

**FIG 5** Representative examples of the different modes of heterochromatin transitions between the active and silenced states of CVGs. (A to I) Representative examples of heterochromatin distribution [log$_2$(H3K9me3/input)] associated with the active or silenced state of CVGs from the different clusters shown in Fig. 4. The genes are representative of "global transition" (cluster 1 [A to C]), "mainly 5′ transition" (cluster 2 [D and E]), "localized 5′ transition" (cluster 3 [F] and cluster 5 [G]), or "mainly 3′ transition" (cluster 6 [H and I]). Panels C and E are representative of transitions that involve the *de novo* formation/complete removal of a heterochromatin domain, whereas the other panels are representative of transitions that can be explained by heterochromatin expansion/retraction. (J and K) Representative examples of heterochromatin distribution in subclones that have a gene in a different transcriptional state and subclones that have it in the same transcriptional state. In all panels, the active or silenced state corresponds to the genes displayed as black arrows (with names). Other annotated genes are shown as gray arrows. The identifiers of the genes displayed as black arrows are PF3D7_0220800 (A), PF3D7_0300900 (B), PF3D7_1302100 (C), PF3D7_0713300 (D), PF3D7_1253000 (E), PF3D7_1039700 (F), PF3D7_0114100 (G), PF3D7_1480000 (H), PF3D7_0936700 (I), PF3D7_0302200 and PF3D7_0302500 (J), and PF3D7_1301500 and PF3D7_1301600 (K). (L) Schematic representation of the four main modalities of heterochromatin transitions between the active and silenced states of CVGs: the *de novo* formation of heterochromatin or the complete disruption of a heterochromatin domain (*De novo* formation/complete removal); the expansion/

While for most genes activation or silencing could be explained by heterochromatin expansion or retraction across a continuous region or small localized transitions, there were 13 H3K9me3 differential peaks in which the *de novo* formation or the complete removal of a heterochromatin domain actually appeared to occur (Fig. 5C and E and Data Set S5). This involved heterochromatin islands isolated from other heterochromatic regions in which, in the active state, the region occupied by these islands and the adjacent regions (at least 5 kb on each side) were fully euchromatic. Unexpectedly, the majority of the 20 genes embedded in these regions were expressed at very low levels both in subclones in which they were euchromatic and in subclones in which they were heterochromatic, and only 5 of them showed differential expression among the 5 subclones (i.e., they were included in the high-confidence list of 121 differentially expressed genes). Indeed, most of these 20 genes were previously reported to be predominantly expressed in gametocytes or mosquito stages rather than during the IDC (Data Set S5). Together, these results suggest that variant gene expression linked to the adaptation of asexual blood-stage parasites to fluctuating conditions during the IDC is regulated mainly by heterochromatin expansion and retraction, whereas the *de novo* formation or the complete removal of heterochromatin is largely restricted to heterochromatin islands containing genes expressed at other stages of development.

**The heterochromatin distribution associated with the active or silenced state of a specific CVG is generally conserved.** For most CVGs, the analysis included more than one subclone in which the gene was either active or silenced. To compare the distribution of heterochromatin between subclones in which the gene was in the same transcriptional state, we first classified all genes in the five subclones as CVGs or non-CVGs and as active or silenced/inactive based on our transcriptomic data (Cy5 signal and expression differences between subclones) and publicly available transcriptome sequencing (RNA-seq) data sets (see Materials and Methods). Genes for which the active or silenced state could not be unambiguously determined were classified as "undetermined" (Data Set S6). In the majority of CVGs, the distribution of heterochromatin was almost identical among subclones in which the gene was in the same transcriptional state (Fig. 5J, Fig. S5, and Data Set S2). In the few genes in which prominent heterochromatin differences were observed between subclones that had the gene in the same state, differences typically occurred toward the end of the coding sequence or downstream region and could be explained by a different transcriptional state of the neighboring genes. As an example, in the active state, *eba-140* had heterochromatin in the second half of the coding sequence and downstream region only when the downstream *phist* gene was silenced (Fig. 5K).

**Heterochromatin differences between the active and silenced states in different families of CVGs.** None of the clusters described in Fig. 4 were strictly linked to a specific CVG family, but different families were unevenly represented among the clusters. Different scenarios were observed for different gene families (Fig. S6). In the *rif*, *stevor*, *pfmc-2tm*, and *var* families, the majority of genes that did not show transcriptional differences among the five subclones analyzed were fully heterochromatic, whereas in the *clag*, *acs*, *hyp*, and *phist* families, nondifferentially expressed genes were typically euchromatic. In some gene families, the majority of differentially expressed genes showed similar heterochromatin transition patterns between the active and silenced states (e.g., localized 5′ transition in the *pfmc-2tm* family), but in other families, the transition patterns were rather heterogeneous.

In specific genes for which we had previously characterized the heterochromatin

**FIG 5** Legend (Continued)

retraction of heterochromatin at the boundaries of a heterochromatin domain [Expansion/Retraction (limits)]; the opening of heterochromatin within a heterochromatin domain, followed by heterochromatin retraction, or the expansion resulting in a continuous heterochromatin domain from two separate neighboring domains [Expansion/Retraction (internal)]; or the opening/closing within a heterochromatin domain with minimal or no expansion/retraction (Localized closing/opening). Changes associated with silencing and activation are indicated by red and green arrows, respectively.

distribution associated with the active or silenced state in the 10G and 1.2B subclones using chromatin immunoprecipitation-quantitative PCR (ChIP-qPCR) (45), the results were fully consistent. In the mutually exclusively expressed *clag3.1* and *clag3.2* genes (26, 45, 73), a heterochromatin domain centered at the *var* pseudogene located between the two genes expanded toward one gene or the other (Fig. 5J). In *eba-140*, the active state was associated with the absence of heterochromatin in the upstream region and the beginning of the coding sequence, whereas heterochromatin in the downstream region did not influence the expression of the gene and reflected the silencing of the neighboring *phist* gene, as described above (Fig. 5K).

**Multiple *var* genes are simultaneously active in some subclones.** Genes of the *var* family typically show mutually exclusive expression such that individual parasites express only one gene of the family at a time and keep all of the others repressed (19–21). To obtain a semiquantitative estimate of the expression patterns of *var* genes in the five 3D7 subclones, we used the normalized Cy5 signal (i.e., not relative to the reference pool) (Fig. 6A), as we did in previous studies (18, 58). Subclones 1.2B and B11 showed the expected pattern of mutually exclusive *var* gene expression, with a single *var* gene being predominantly expressed, whereas no expressed *var* gene was identified in 10G. Unexpectedly, subclones A7 and E5 expressed several *var* genes (four and six, respectively) at relatively high levels.

Two different scenarios can explain this observation: (i) authentic simultaneous expression of multiple *var* genes in individual parasites or (ii) transcriptional heterogeneity such that despite the use of recently subcloned lines, different subsets of parasites express different *var* genes. To distinguish between these two possibilities, we used the H3K9me3 ChIP-seq data. If each *var* gene was active in only a subset of the parasites, a partial reduction of heterochromatin levels would be expected, but heterochromatin was almost completely depleted at the regulatory regions of all but one (PF3D7_0413100 in A7) of the highly expressed *var* loci (Fig. 6B). This result suggests that multiple *var* genes were actually simultaneously active in the A7 and E5 subclones. The majority of the *var* genes that were expressed at high levels in one or more of the subclones and silenced in the others showed major differences between the active and silenced states (cluster 1, 2, or 6 in Fig. 4), although two showed more localized differences (cluster 3) (Data Set S2). While an unambiguous demonstration of the simultaneous expression of multiple *var* genes in the same cell requires analyses at the single-cell level, such as RNA fluorescence *in situ* hybridization (RNA-FISH) or single-cell RNA-seq (scRNA-seq), and we did not determine if the multiple *var* genes in an active chromatin state produce full-length mRNA and PfEMP1 protein, our results are consistent with previous reports indicating that the mutually exclusive expression of *var* genes may not be strict (74, 75).

**Differences in heterochromatin distribution between parasite lines with different sexual conversion rates.** The proportion of parasites that convert into gametocytes at each round of the IDC is called the sexual conversion rate. It is well established that different parasite lines, and even different subclones of the same genetic background, have different rates of basal (noninduced) sexual conversion that are inherited across multiple generations of asexual blood growth (34, 76). Sexual conversion depends on the activation of *pfap2-g*, which is normally silenced by heterochromatin in asexually growing blood-stage parasites. This gene is a peculiar CVG because, unlike other CVGs, only the silenced state can be transmitted from one generation of the IDC to the next: parasites in which the gene is activated convert into nonreplicating gametocytes and abandon the IDC. Therefore, the stable differences in the conversion rates between parasite lines imply the transmission of the probability of activation of *pfap2-g* rather than the transmission of an active or silenced state. To identify potential epigenetic determinants of the probability of activation of *pfap2-g*, we compared the distribution of heterochromatin among the five 3D7 subclones and additional parasite lines with a broad range of sexual conversion rates (0 to >10%) (Fig. 1A). All parasite lines were maintained in standard culture medium containing Albumax II and no human serum, with the exception of NF54-V63 (previously published data), which was analyzed in medium containing human serum (58).

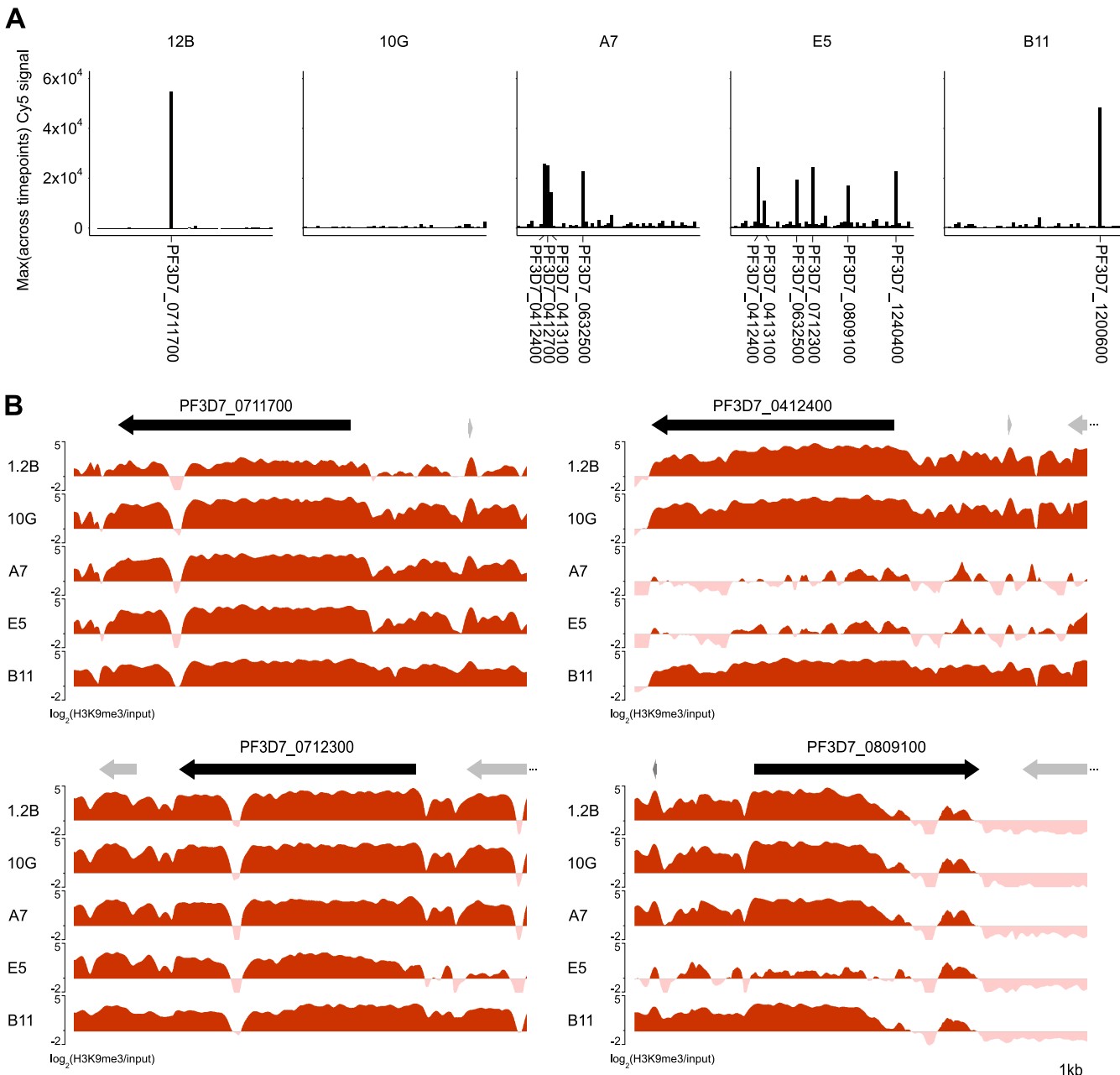

**FIG 6** Expression patterns and heterochromatin distribution at *var* gene loci. (A) Expression of *var* genes in the five subclones. Values are the normalized sample signal (Cy5 channel), which provides a semiquantitative estimation of the absolute expression levels. The gene identifier is provided only for *var* genes expressed at high levels (normalized Cy5 signal intensity of >10,000). (B) H3K9me3 distribution at representative *var* genes that are highly expressed in some of the subclones. Black arrows (with identifiers) are the differentially expressed *var* genes; other annotated genes are shown as gray arrows. Values are log$_2$(H3K9me3/input).

The heterochromatin occupancy at the *pfap2-g* locus was similar among all of the parasite lines, with similar 5′ and 3′ boundaries of the H3K9me3-enriched region (Fig. 7). Therefore, a different extension of the heterochromatin domain at the *pfap2-g* locus does not appear to underlie the different probabilities of *pfap2-g* activation between parasite lines. There were subtle differences between parasite lines in the H3K9me3 coverage at the 5′ and 3′ ends of the H3K9me3 domain, but these may be explained by the different proportions of parasites that were sexually committed (and therefore had *pfap2-g* in an active state) rather than authentic epigenetic differences transmitted across multiple rounds of the IDC.

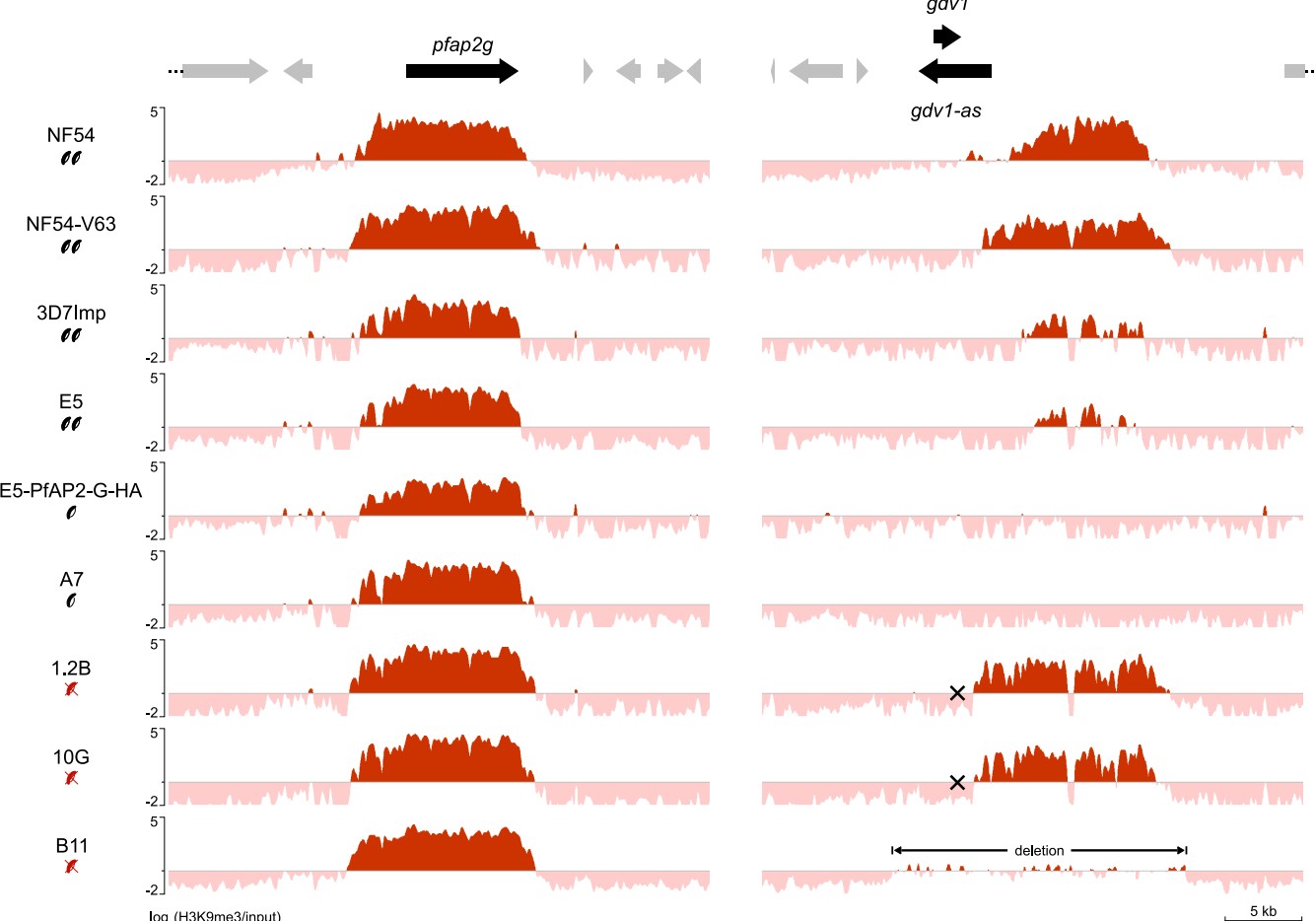

**FIG 7** Distribution of heterochromatin at the *pfap2-g* and *gdv1* loci encoding regulators of sexual conversion. The distribution of H3K9me3 at the *ap2-g* and *gdv1* loci in different parasite lines is shown. Values are log₂(H3K9me3/input). Parasite lines are semiquantitatively classified as high-gametocyte producers (two gametocytes, >10% sexual conversion rate), low-gametocyte producers (one gametocyte, <3% sexual conversion rate), or non-gametocyte producers (red, crossed gametocytes), as described in the legend of Fig. 1. Mutations that produce a premature stop codon in *gdv1* (crosses) or a deletion including the full *gdv1* locus are indicated. Other annotated genes are indicated as gray arrows.

In contrast, we observed major differences between the parasite lines in the heterochromatin distribution at the locus encoding GDV1, an upstream positive regulator of *pfap2-g* (69, 70, 77) (Fig. 7). While *gdv1* is typically not considered a CVG or a heterochromatic gene, in some of the parasite lines, we observed a prominent H3K9me3 signal downstream of the coding sequence. This heterochromatin domain overlaps the putative promoter of the long noncoding antisense RNA (*gdv1-as*) that was shown previously to operate as a negative regulator of GDV1 (69) and other previously reported long noncoding RNAs located in this large intergenic region (78). Of note, parasite lines with high (>10%) sexual conversion rates (NF54, NF54-V63, 3D7-Imp, and E5) (34, 58, 66) tended to have prominent heterochromatin at this position, whereas heterochromatin was absent from parasite lines with low (<3%) conversion rates (A7 and E5-PfAP2-G-HA, the latter derived from E5 but with a much lower conversion rate after genome editing and subcloning) (34, 65). The exceptions were the parasite lines 1.2B and 10G, which had prominent heterochromatin at the *gdv1* locus despite not producing gametocytes. However, these subclones have a nonsense mutation that results in a premature stop codon in *gdv1* (Data Set S2) (35) that likely makes the protein nonfunctional and uncouples the regulation of the *gdv1* locus from sexual conversion. Another study also showed that the premature truncation of GDV1 makes the protein nonfunctional (79). The other non-gametocyte-producer line in our study, B11, has a large deletion that spans the *gdv1* coding sequence and the downstream

region (Fig. S1). Together, these results suggest that heterochromatin at the *gdv1* locus plays a role in the regulation of sexual conversion rates, possibly by affecting the expression of *gdv1-as*.

**Changes in H3K9ac associated with the transcriptional state of CVGs.** We also characterized the distribution of H3K9ac in the 10G, 1.2B, and B11 subclones by ChIP-seq as an exploratory analysis to determine if, in combination with H3K9me3, analysis of this mark may help to identify specific signatures for active and silenced CVGs. H3K9ac is typically associated with the active state of CVGs (45–48) and is also abundant at constitutively euchromatic genes, mainly in intergenic regions (53, 54). There was high overall similarity in the distribution of H3K9ac between subclones (mean pairwise correlation using input-normalized values of 0.92 [range, 0.90 to 0.96]) (Fig. S2A), and the distribution of H3K9ac was largely nonoverlapping with that of H3K9me3, as previously described (51, 54) (Fig. 8A and B). Active CVGs typically had H3K9ac in the region depleted of H3K9me3 (Fig. 8B). We also generated H3K9ac ChIP-seq data for the parasite lines 3D7-Imp, NF54, and E5-PfAP2-G-HA (available in the GEO database), but these data were not further analyzed because transcriptomic analysis was not performed for these parasite lines.

To compare the distribution of H3K9me3 and H3K9ac between the active or silenced CVGs and the active or inactive non-CVGs, we used the classifications of the genes in these different categories for each subclone (Data Set S6). We compared the coverages of H3K9ac and H3K9me3 in the upstream (1,000 bp), coding, and downstream (1,000 bp) regions between genes in the different categories (Fig. 8C). As expected, H3K9me3 was absent from non-CVGs and more abundant in silenced CVGs than in active CVGs. In contrast, the H3K9ac coverage was higher in non-CVGs than in CVGs. Among the CVGs, the H3K9ac coverage was higher in active CVGs than in silenced CVGs, especially in the upstream and coding regions. In these regions, the H3K9ac coverage in active CVGs was almost as high as that in non-CVGs. We also compared the ratios of H3K9ac to H3K9me3, which revealed a pattern similar to that for H3K9ac levels.

To determine if genes in the different categories showed distinct H3K9 signatures, we generated two-dimensional scatterplots with H3K9me3 and H3K9ac coverages at the upstream, coding, and downstream regions (Fig. 8D). Both non-CVGs and silenced CVGs formed compact clusters, whereas the active CVG cluster was more scattered and overlapped the two other clusters. Thus, the global levels of H3K9 epigenetic marks alone cannot be used to fully discriminate between non-CVGs and active CVGs or between active and silenced CVGs. Instead, algorithms that consider the distribution of H3K9 marks associated with the active or silenced state of each specific CVG will be needed to predict the transcriptional state of a CVG from epigenomic data.

## DISCUSSION

Changes in the expression of CVGs underlie several important processes in malarial host-parasite interactions, including antigenic variation, alteration of infected RBC cytoadherence tropism and solute permeability, the use of alternative invasion pathways, and sexual conversion (13, 14, 17). Transitions between the heterochromatic and euchromatic states at CVG loci drive expression switches, but the heterochromatin distribution associated with the active and silenced states has been previously characterized for only a few genes (45–48). Here, we systematically characterized the heterochromatin distribution associated with the active and silenced states of over a hundred CVGs, which revealed diverse modalities of heterochromatin changes associated with transcriptional switches. Of note, all of the characterized expression switches and heterochromatin changes occurred during the IDC because all of the subclones were derived from the clonal 3D7 line and had not gone through mosquito passage since the 3D7 line was established. In the majority of the 121 genes for which we compared the distribution of heterochromatin between the active and silenced states, the transitions can be explained by heterochromatin retraction and expansion. Heterochromatin spreading was previously demonstrated to occur in *P. falciparum* (73). Therefore, we propose that CVG expression

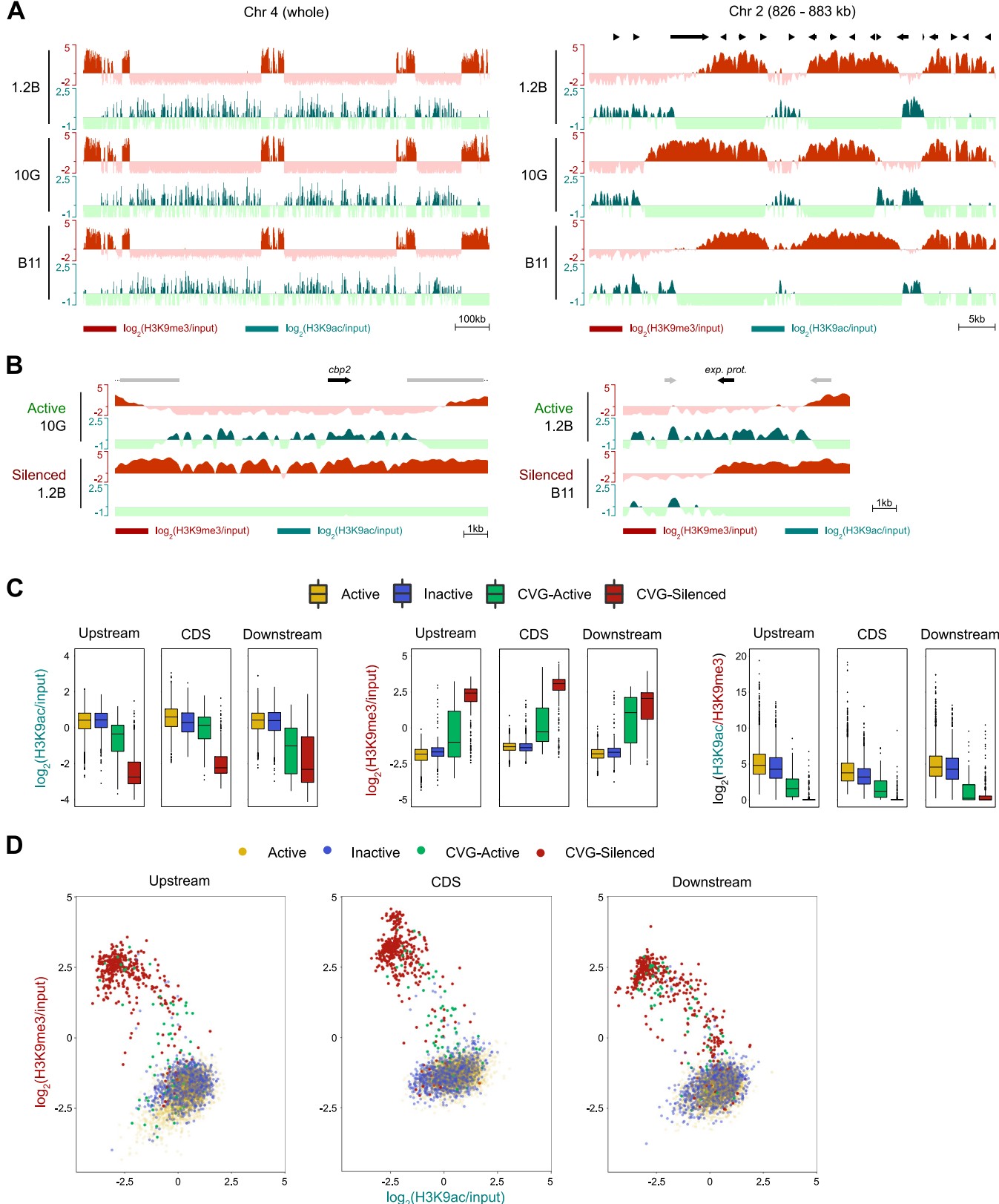

**FIG 8** Distribution of H3K9ac and H3K9me3 in CVGs and non-CVGs. (A) General distribution of H3K9ac and comparison with the distribution of H3K9me3 in a representative chromosome or chromosomal region. (B) Representative examples of changes in H3K9me3 and H3K9ac associated with the active or silenced states of CVGs. The identifiers of the genes displayed as black arrows are PF3D7_1301700 (left) and PF3D7_0532600 (right). Other annotated genes are indicated as gray arrows. (C) Mean coverage of H3K9ac and H3K9me3 and H3K9ac/H3K9me3 ratios at the upstream region (1,000 bp), the coding

during the IDC is typically regulated by an accordion-like mechanism rather than by the *de novo* formation or the complete removal of heterochromatin domains (Fig. 5L). The latter mechanism appears to be common only for genes that are expressed at other stages of the life cycle (i.e., transmission stages). The significance of heterochromatin at these genes for expression during transmission stages is not known.

A key aspect of the design of our study was the use of subclones that had been cultured for a limited number of generations after subcloning, which ensures that they were largely homogeneous for the transcriptional state of the majority of the CVGs. In parasite lines that have not been recently subcloned, different individual parasites have different combinations of expressed and silenced CVGs (18), and this heterogeneity precludes the detailed characterization of the active and silenced states of the genes. Other important aspects of our study design were comparing subclones of the same genetic background and combining genome-wide epigenomic analysis at high resolution (using small chromatin fragments) with transcriptomic analysis across the full IDC of the same parasite lines.

Because the *P. falciparum* genome is compact, with relatively small intergenic regions, the distribution of heterochromatin in one CVG locus was often influenced by the state of neighboring CVGs. This likely contributes to the complexity and variety of patterns of heterochromatin differences observed between the active and silenced states. For example, in a gene that has a heterochromatic downstream neighbor, the end of the coding sequence is typically heterochromatic even in the active state. Despite this, CVGs are regulated as individual units such that a CVG can be activated while its neighboring CVGs remain silenced. This suggests the existence of boundary elements, such as barrier insulators (80, 81), that prevent the spreading of heterochromatin beyond certain sequence elements. Based on the observation that there are positions beyond which heterochromatin never spreads, neither in the parasite lines analyzed here nor in parasite lines of different genetic backgrounds (52), we speculate that there may be "strong" barrier elements that strictly prevent spreading into euchromatic areas containing essential non-CVGs. Additionally, there are specific positions where heterochromatin spreading commonly stops in different subclones that have a CVG in the active state but not in subclones in which the gene is silenced (heterochromatin domain limits are generally conserved among subclones with a CVG in the same state). This may reflect the presence of "weak" barrier elements that enable the stable activation of a CVG amid heterochromatic regions but can eventually be overcome, resulting in a switch toward the silenced state.

The phenotype of a parasite depends not only on its genome sequence but also on how it uses its CVGs. In natural human malaria infections, thousands of full parasite genomes have been sequenced (82, 83), but characterizing transcriptomes across the full IDC is far more complex (16). The main difficulty in studying *P. falciparum* gene expression in human infections is that only ring-stage parasites and mature gametocytes are found in the circulation, whereas all other blood stages are sequestered in different organs or tissues and therefore are not available in peripheral blood samples. Consequently, the expression of genes such as *var* or *pfap2-g*, which are active in ring stages, can be directly characterized in human blood samples (20, 84–89), but the expression of genes expressed at other stages of the IDC cannot. In studies using genome-wide transcriptome analysis of patient samples, only the expression of genes expressed in rings is captured (16, 84, 90, 91). Some full-transcriptome studies (58, 92, 93) and studies focused on specific CVGs such as *clag* (60) or invasion genes (31, 94, 95), all expressed in late stages of the IDC, used *ex vivo* parasite cultures to obtain samples for transcriptional analysis at different stages of the IDC. However, this approach is

**FIG 8** Legend (Continued)

sequence (CDS), and the downstream region (1,000 bp) of genes classified as being (non-CVG) active, (non-CVG) inactive, CVG active, or CVG silenced. All genes from the 1.2B, 10G, and B11 subclones assigned to one of these categories were included in the analysis. Boxes are interquartile ranges (with medians), and whiskers indicate the 5th and 95th percentiles. (D) Scatterplot of H3K9ac versus H3K9me3 coverage for genes classified as being active, inactive, CVG active, or CVG silenced.

time-consuming and requires culture facilities at the place of sample collection, which compromises its feasibility. Together, these difficulties limit our capacity to investigate CVG expression under different clinical presentations, in hosts of different ages, or under different epidemiological conditions.

The distribution of heterochromatin does not change during the full IDC (45–47, 52), implying that H3K9me3 or PfHP1 ChIP-seq analysis performed at a single time point of the IDC informs about the distribution of heterochromatin for CVGs expressed at any asexual blood stage. Since we found that the distribution of heterochromatin is generally conserved among different parasite lines that have a CVG in the same transcriptional state, it is reasonable to expect that it should be possible to predict the transcriptional state of a CVG across the full IDC from the heterochromatin distribution at a single stage (e.g., the ring stage available in peripheral blood samples). However, the influence of the state of neighboring CVGs needs to be taken into consideration because for some genes, this influence results in more than one possible heterochromatin distribution associated with the same transcriptional state of a gene. We envisage that algorithms using machine learning or other approaches can be developed to accurately predict in human infections the expression state of essentially all CVGs from H3K9me3 ChIP-seq analysis of circulating rings. While developing such classification algorithms is clearly beyond the aim of our study, the data set presented here could be used as a seed for this type of algorithm, which later on would be improved with further data sets as they become available. We are aware that chromatin extraction and ChIP-seq are technically more demanding than RNA extraction and microarray or RNA-seq analysis, but this would clearly be compensated for by avoiding the need for *ex vivo* culture and sampling at multiple times to capture the full IDC. Furthermore, new technical developments such as cleavage under targets and tagmentation (CUT&Tag) (96), which uses streamlined protocols and requires a small amount of sample, may facilitate heterochromatin profiling directly from human infections. In addition to H3K9me3, many other histone posttranslational modifications have been described in *P. falciparum* (54, 97–99), but the majority are a consequence, rather than a cause, of the transcriptional state of a gene, or they fluctuate as the parasite progresses along the IDC. In contrast, H3K9me3-based heterochromatin actually determines the state of CVGs, and its distribution is stably maintained across the full IDC over multiple generations, making it a truly epigenetic mark that carries heritable information (100). Therefore, profiling H3K9me3 (or PfHP1) may be sufficient to predict the transcriptional state of most CVGs. Additionally, profiling H3K9ac may also be informative, especially when a CVG is active in only a small fraction of the parasites.

In addition to the global analysis of heterochromatin changes at CVGs, we paid special attention to specific genes with important known functions in host-parasite interactions. We found that in some subclones, several *var* genes were simultaneously in an active chromatin state (and expressed at high levels) in the majority of parasites in the population. These results suggest that the mutually exclusive expression of *var* genes may not be strict, consistent with previous studies that reported the expression of multiple *var* genes in a single parasite after selection for binding to multiple receptors (74) or the expression of several *var* genes in some recently subcloned lines (75). As an exploratory analysis, we also investigated epigenomic traits that may correlate with sexual conversion rates, a phenotype for which the molecular basis of variation has not been established. This resulted in the identification of a heterochromatin domain at the *gdv1* locus as a candidate regulator of sexual conversion rates. Heterochromatin was present at the putative promoter of the *gdv1-as* long noncoding RNA in parasite lines with high sexual conversion rates or lacking functional GDV1, but not in parasite lines with low sexual conversion rates and an intact *gdv1* gene. This result suggests that heterochromatin at this locus may prevent the expression of *gdv1-as*, which may enable the expression of GDV1 (69) and the subsequent activation of the master regulator of sexual conversion, *pfap2-g*. However, further studies will be needed to test this hypothesis, including analyses under defined conditions that induce or repress sexual conversion (9, 101) and manipulation of the locus using transfection approaches.

## MATERIALS AND METHODS

**Parasites.** The 3D7-A and 3D7-B stocks of the *P. falciparum* clonal line 3D7; the 3D7-A subclones 10G and 1.2B; the 3D7-B subclones A7, E5, and B11; and the transgenic E5-PfAP2-G-HA line (9A subclone) were previously described and characterized (18, 26, 34, 61, 62). The 3D7-Imp line (66, 102) is the stock of 3D7 (103) maintained at Imperial College. The NF54 line (103) was from the stock maintained at Sanaria (104), and the NF54-V63 line was obtained after infecting a volunteer with Sanaria cryopreserved NF54 sporozoites in a CHMI trial (58, 105). Cultures for all of the new experiments presented in this study (and for the previous transcriptomic analysis of 1.2B, 10G, and 3D7-B [18]) were maintained at 3% hematocrit with $B^+$ RBCs in standard parasite culture medium with Albumax II and no human serum in an atmosphere containing 5% $CO_2$ and 3% $O_2$, balanced with $N_2$. However, the NF54-V63 line (for which ChIP-seq data generated as part of a previous study are included in some analyses) was cultured in medium containing human serum (58). Sexual conversion rates were determined using *N*-acetyl-D-glucosamine treatment and light microscopy, as previously described (65).

**Transcriptomic experiments.** Samples for transcriptomic analysis of the A7, E5, and B11 lines were obtained from tightly synchronized cultures (5-h age window) obtained by Percoll purification of RBCs infected with mature stages followed by 5% sorbitol lysis 5 h later. RNA was extracted from samples collected at 10 to 15, 20 to 25, 30 to 35, 37 to 42, 40 to 45, and 43 to 48 h postinvasion (hpi) using the TRIzol method, as previously described (18, 106). After reverse transcription and labeling, Cy5-labeled samples were hybridized against a Cy3-labeled reference pool, as previously described (68). The reference pool consisted of a mixture of equal amounts of cDNA from rings, trophozoites, and schizonts from 3D7-A and 3D7-B, similar to the reference pool used for the analysis of 1.2B, 10G, and 3D7-B but from a different biological preparation (18). Samples were hybridized on previously described custom Agilent microarrays (AMADID no. 085763) (35) modified from the AMADID no. 037237 design (68).

**ChIP-seq experiments.** Samples for ChIP-seq analysis were obtained from sorbitol-synchronized cultures at the late trophozoite/early schizont stage. Since the heterochromatin distribution is stable during the IDC, we did not use cultures tightly synchronized to a 5-h age window for these experiments. Chromatin extraction and immunoprecipitation were performed essentially as previously described (35, 58). In brief, after saponin lysis and formaldehyde cross-linking, chromatin was extracted using the MAGnify chromatin immunoprecipitation system (Life Technologies). Chromatin was sonicated using a Covaris M220 sonicator to obtain fragments of ~150 to 200 bp. For immunoprecipitation, we used 4 $\mu$g of chromatin and 8 $\mu$g of antibodies against H3K9me3 (catalog no. C15410193; Diagenode) or H3K9ac (catalog no. C15410004; Diagenode). For library preparation, we used 4 ng of input or immunoprecipitated DNA, the NEBNext multiplex oligonucleotides for Illumina (New England BioLabs [NEB]), 8 to 10 cycles of amplification with the Kapa HiFi PCR kit (Kapa Biosystems), and AMPure XP beads for purification steps (35, 107). Sequencing was performed using an Illumina HiSeq 2500 system, obtaining 8 million to 39 million 125-bp paired-end reads per sample.

**Microarray data analysis.** For the 1.2B, 10G, and 3D7-B lines, we used previously published processed expression values (18). For the A7, E5, and B11 subclones, microarray data were analyzed similarly, but normalized Cy5 and Cy3 values were obtained directly using Agilent Feature Extraction software. The rest of the analysis was performed using mainly the Bioconductor and the Tidyverse suite libraries in an R environment (v.3.6.3), essentially as previously described (58). In brief, for each channel, the background signal intensity was calculated as the median signal of the 100 lowest-signal probes in each array. Probes with a signal for both channels below three times the background were excluded from further analysis. For each probe, the expression value was defined as the $\log_2$(Cy5/Cy3) signal. Probes were collapsed into genes using median polish. For each array, the estimated parasite age in hours postinvasion was calculated using a previously described maximum likelihood method (67). For each gene and subclone, expression plots were generated with gene expression values on the *y* axis and estimated hours postinvasion on the *x* axis. These plots were used to calculate the average fold change (AFC) for each gene over four overlapping time intervals of half the estimated parasite age difference (in hours postinvasion) between the first and last analysis time points. The AFC was calculated for all possible pairwise comparisons among A7, E5, and B11 according to a previously described pipeline (18). The maximum AFC (mAFC) was defined as the AFC at the time interval in which it had the highest absolute value.

From this point, data from 1.2B, 10G, 3D7-B, A7, E5, and B11 were analyzed in parallel, although comparisons were performed separately for 1.2B, 10G, and 3D7-B or for A7, E5, and B11 (because these two groups of samples were analyzed with slightly different microarray platforms and reference pools). Genes with an mAFC of >4 (in at least one of the pairwise comparisons among 1.2B, 10G, and 3D7-B or among A7, E5, and B11) were considered to have markedly different transcript levels. To identify differentially expressed genes with high confidence, we applied several filters, essentially as previously described (58). For each pairwise comparison, only genes that had an mAFC of >4 and passed all of the filters were included in the final list of differentially expressed genes. First, tRNAs and genes lacking a PlasmoDB identification were excluded. Second, genes with very low absolute expression values were excluded. For this, genes with a Cy5 signal within the lowest 15% (at all time points) in the parasite lines with an mAFC of >4 in the pairwise comparison were filtered out. Third, genes that showed large expression differences only at the time periods in which they were expressed at low levels were excluded. For this, genes that at the time interval of maximum expression did not have an AFC of >4 were excluded. Finally, genes (as annotated in PlasmoDB v52, including the 5′ UTR and 3′ UTR when available) that overlapped a duplication or a deletion (see below) in at least one of the parasite lines among which the expression difference was observed were also excluded. To classify genes as previously reported CVGs, we used a previously described list of CVGs (58), whereas for the identification of gametocyte markers (genes with higher expression levels in sexual than in asexual parasites), we used a

list based on previously reported transcriptomic studies of gametocytes (see Data Set S7 in the supplemental material).

For some analyses of differentially expressed genes, we focused on the pairwise comparison with the largest fold difference in expression to identify representative subclones for the silenced and active states. For this, for each gene, we selected the pairwise comparison that passed all filters and had the largest mAFC among the comparisons of 1.2B versus 10G, 1.2B versus 3D7-B, 10G versus 3D7-B, A7 versus E5, A7 versus B11, and E5 versus B11. When the largest mAFC occurred between 3D7-B (which is not a transcriptionally homogeneous subclone and for which we did not generate ChIP-seq data) and 1.2B or 10G, we first determined which 3D7-B subclone (A7, E5, or B11) had the maximum (when the gene was active in 3D7-B and silenced in 10G or 1.2B) or minimum (when the gene was silenced in 3D7-B and active in 10G or 1.2B) expression value and selected this subclone as the representative subclone for the active or silenced state of the gene.

**ChIP-seq data analysis.** ChIP-seq data analysis, including the generation of plots, was performed using custom Python (v.3.6.9) and R (v.3.6.3) scripts, the latter mainly using Tidyverse suite packages. The initial processing of the data was performed approximately as described previously (35). After a quality check using FastQC (v.0.11.9), reads were trimmed, and repetitive $k$-mers were removed using BBDUK (v.36.99) with the parameters ktrim = r, k = 22, and mink = 6. Reads were aligned to the reference genome (*P. falciparum* 3D7 [PlasmoDB v28]) using Bowtie2 (v.2.3.0), and duplicate reads were excluded using the PICARD suite (v.4.1.9.0) with the RemoveDuplicates command. Peak calling was performed using MACS2 (v.2.2.7.1) with the parameters -f BAMPE -B -g 2.41e7 –keep-dup all –fe-cutoff 1.5 -nomodel –extsize 150. To calculate the normalized coverage of enrichment over the input ($log_2$ transformed) at a 10-bp resolution, we used the DeepTools (v.3.5.0) BamCompare command with the parameters –normalizeUsing RPKM -bs 10 –smoothLength 200 –pseudocount 10. After setting negative coverage values to zero to prevent the detection of coverage differences in nonheterochromatic regions, the coverage for landmark regions (e.g., upstream, coding, or downstream sequences of genes) was calculated as the mean value for the 10-bp bins within the region using BedTools (v.2.26.0). Custom scripts were used to create BED files demarcating the regions of interest for each gene for each particular analysis.

To identify differential H3K9me3 peaks in all possible pairwise comparisons among subclones 10G, 1.2B, E5, A7, and B11, we used custom scripts. First, the input-normalized coverage was calculated for 100-bp bins in all regions overlapping a MACS2 peak in at least one of the parasite lines of the pairwise comparison. The distribution of coverage intensities was adjusted to a skewed normal distribution, and for each bin, we calculated a cumulative density function (CDF) value for each parasite line. Bins with a CDF difference of >0.3 were considered to have differential coverage. Next, bins with differential coverage separated by <500 bp were merged, and differential peaks of <1,000 bp after merging were filtered out.

**Identification of genetic changes in the parasite lines used in this study.** To identify genetic polymorphisms, we analyzed the sequences from the input samples of the ChIP-seq experiments. To identify single nucleotide polymorphisms (SNPs) and short indels, we used the variant-calling protocol in the GATK suite according to GATK best practices (Data Set S2). Mutations with a coverage of <20 reads were excluded. Since recently subcloned lines are genetically highly homogeneous, a very high prevalence is expected for real mutations; therefore, we retained only mutations (relative to the reference 3D7 genome in PlasmoDB v52) in which the frequency of the reference allele was ≤0.5. Mutations were annotated using the ENSEMBL variant effect predictor (VEP). To define regions with relatively large (>500-bp) duplications or deletions, we used custom scripts. We first calculated coverage values for the inputs of the ChIP-seq data using the DeepTools (v.3.5.0) BamCoverage command with the parameters –normalizeUsing RPKM -bs 10 –smoothLength 200, similar to the calculation of the coverage for input-normalized ChIP-seq values. For each sample, the mean coverage over the whole genome was calculated. Bins with a coverage value <0.05 times the mean coverage were marked as underrepresented, and bins with a coverage value >1.75 times the mean coverage were marked as overrepresented. Bins marked as underrepresented or overrepresented that were separated by <200 bp were joined, and the resulting fragments were discarded if they were <500 bp. Finally, to avoid marking as deletion regions of low coverage due to the underlying sequence, we removed all regions with putative deletions that were common to all of the subclones analyzed. The remaining fragments were marked as deletions or duplications.

**Classification of genes according to their transcriptional state.** In order to assign a transcriptional state to each gene in subclones 1.2B, 10G, A7, E5, and B11, we designed a tree-like classification system. The final possible categories were active, inactive, undetermined, CVG active, CVG silenced, and CVG undetermined. The classification system is illustrated in Fig. S7, and the state of each gene is provided in Data Set S6.

The main parameters used for the classification of genes were: (i) previous classification as a CVG on the basis of variant expression or the presence of heterochromatin marks, according to a previously published list of CVGs (58); (ii) differential expression among the five parasite subclones in the data set presented here, with an mAFC of >2 at the time of maximum expression; (iii) Cy5 (sample channel) value percentiles in our data set; and (iv) expression level percentiles in published RNA-seq data sets (53, 107–109). In brief, genes were first classified as CVGs or non-CVGs. Next, regardless of whether a gene was a CVG or not, genes that showed no expression differences among the subclones were classified as active or inactive/silenced according to their maximum expression percentiles, whereas genes with expression differences were classified as active or inactive/silenced according to their relative expression levels. Genes with intermediate expression levels, duplications, or deletions were classified as undetermined or CVG undetermined.

**Availability of data.** The new microarray and ChIP-seq data described in this article have been deposited in the GEO database with accession no. GSE208561. ChIP-seq data are also available for visualization

at the UCSC genome browser (https://genome.ucsc.edu/s/lucas_michel_todo/3D7A%20and%203D7B %20subclones%20H3K9me3%20and%20H3K9ac and https://genome.ucsc.edu/s/lucas_michel_todo/NF54 _3D7imp_E5HA). All of the scripts used for the data analysis are available at GitHub (https://github.com/ LucasMichelTodo/Heterochomatin_Transitions.git).

## SUPPLEMENTAL MATERIAL

Supplemental material is available online only.
**SUPPLEMENTAL FILE 1**, PDF file, 2.4 MB.
**SUPPLEMENTAL FILE 2**, XLSX file, 1.1 MB.
**SUPPLEMENTAL FILE 3**, XLSX file, 0.2 MB.
**SUPPLEMENTAL FILE 4**, XLSX file, 0.1 MB.
**SUPPLEMENTAL FILE 5**, XLSX file, 0.6 MB.
**SUPPLEMENTAL FILE 6**, XLSX file, 0.1 MB.
**SUPPLEMENTAL FILE 7**, XLSX file, 0.4 MB.
**SUPPLEMENTAL FILE 8**, XLSX file, 0.02 MB.

## ACKNOWLEDGMENTS

We are grateful to Mariona Bustamante (ISGlobal) for help in the acquisition of funding, support in setting up the project, and critical reading of the manuscript; S. Hoffman (Sanaria) for providing the NF54 line; and M. Delves (Imperial College) for providing the 3D7-Imp line.

This work was supported by grants from the Spanish Ministry of Science and Innovation (MCI)/Agencia Estatal de Investigación (AEI) (SAF2016-76190-R and PID2019-107232RBI00/ AEI/10.13039/501100011033 to A.C.), cofunded by the European Regional Development Fund (ERDF) (European Union), the la Caixa Banking Foundation (HR18-00267 to A.C.), and the ISGlobal Alliance pilot project program (to C.B., J.R.G., and A.C.). L.M.-T. is supported by a fellowship from the Spanish Ministry of Economy and Competitiveness (BES-2017-081079), cofunded by the European Social Fund (ESF). Our research is part of ISGlobal's Program on the Molecular Mechanisms of Malaria, which is partially supported by the Fundación Ramón Areces. We acknowledge support from the Spanish Ministry of Science and Innovation through the Centro de Excelencia Severo Ochoa 2019–2023 Program (CEX2018-000806-S) and support from the Generalitat de Catalunya through the CERCA Program.

A.C. and C.B. designed the experiments. C.B. optimized and performed ChIP-seq experiments, and N.C.-V. and N.R.-G. performed transcriptomic experiments. L.M.-T. performed the bioinformatic analysis of the data and generated the figures. C.H.-F. and J.R.G. performed preliminary analyses of the initial data. L.M.-T. and A.C. interpreted the results and wrote the manuscript, with input from all authors. A.C. conceived the project. C.B., J.R.G., and A.C. obtained funding.

We declare that we have no competing interests.

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
