## [Reviewer comments · Microbiology Spectrum]

Microbiology Spectrum

Patterns of heterochromatin transitions linked to changes in the expression of *Plasmodium falciparum* clonally variant genes

Lucas Michel-Todó, Cristina Bancells, Núria Casas-Vila, Núria Rovira-Graells, Carles Hernández-Ferrer, Juan Gonzalez, and Alfred Cortés

Corresponding Author(s): Alfred Cortés, Instituto de Salud Global Barcelona

Review Timeline:

Submission Date:	August 4, 2022
Editorial Decision:	October 18, 2022
Revision Received:	November 3, 2022
Accepted:	November 22, 2022

Editor: Laura Kirkman

Reviewer(s): Disclosure of reviewer identity is with reference to reviewer comments included in decision letter(s). The following individuals involved in review of your submission have agreed to reveal their identity: Francesca Florini (Reviewer #1); Jun Miao (Reviewer #3)

Transaction Report:

DOI: <https://doi.org/10.1128/spectrum.03049-22>

October 18, 2022

Prof. Alfred Cortés
ISGlobal
Barcelona
Spain

Re: Spectrum03049-22 (Patterns of heterochromatin transitions linked to changes in the expression of *Plasmodium falciparum* clonally variant genes)

Dear Prof. Alfred Cortés:

Thank you for your patience. Please find the attached comments from two reviews. It appears there are mainly clarifications needed.

Link Not Available

Sincerely,

Laura Kirkman

Journals Department
Reviewer comments:

Reviewer #1 (Comments for the Author):

In this study, Michel-Todó and colleagues compare transcriptomic analysis with heterochromatin distribution in order to assess correlation between gene expression and chromatin accessibility in freshly derived wildtype clones. The analysis by ChIP-Seq represents the novelty compared to the previous study from Rovira-Graells, 2012. Two clones from that work are re-analyzed here. Additionally, three fresh clones from a different 3D7 clones are added in the present work.

It is unclear to me the choice of the authors of re-analyzing only two of the three clones from the previous study. Why for the analysis they include the 3D7-B parental line instead of clone W4-1, which was generated in that study along with clones 1.2B

and 10G? The use of three clones from the previous study versus three clones from the present study would make all the analysis more robust (see for example Figure 3).

This is especially true since the author state that due to different experimental setups they are unable to compare clones from the two studies for the transcriptomic analysis.

If the clone was discarded for any specific reason, the authors should discuss that in the text.

It is anyway my opinion that for the work to be robust enough, the analysis should be done on at least three clones per group of subclones.

Figure 2B and lines 213-214: "ChIP-seq analysis of H3K9me3 revealed a similar global distribution of heterochromatin between the five subclones, although differences were apparent at specific loci". These findings are described too briefly. The authors need to expand the description on what are the loci shown in the figure and what could be the significance of those differences. At least, in the figure, the genes marked with different chromatin status should be more clearly labeled.

Figure 3: why is this analysis performed only on 1.2B vs 10G, and not on the other three new clones?

Paragraph line 380-391: I think the results here are overinterpreted. It is a stretch to make a claim about var genes expression at single parasites by the heterochromatin distribution of the population.

Figure 7: It would be interesting to include in the figure a larger part of the upstream region of *gdv1*, to appreciate that the heterochromatin formation is only in the promoter of *gdv-as* and not in the one of *gdv1*.

Line 445-446: it is again unclear to the reader why only those three clones were analyzed for H3K9ac and not all five available clones.

Reviewer #3 (Comments for the Author):

The submission from Lucas Michel-Todó et al characterized patterns of heterochromatin transitions of CVGs by checking the association between gene expression and heterochromatin mark, H3K9me3 and active mark H3K9ac in different clones of the human malaria parasite *Plasmodium falciparum*. The topic of CVG transition or switch in these parasites has been studied for many years and is critical for understanding this epigenetic mechanism. The results support that expression switches are probably caused by the expansion or retraction of heterochromatin domains. The authors also confirm the presence of multiple active var genes in some clones. Heterochromatin levels in the *gdv-1* ncRNA were positively correlated with the sexual conversion rates. Finally, the H3K9ac landscape was mostly not overlapped with heterochromatin domains. While the results are most convincing and add to the body of work on this topic, some (multiple var gene expression, H3K9ac landscape) are to a large extent confirmatory of previously published work that used similar approaches.

1). While the heterochromatin patterns in Fig. 5 are mostly descriptive without quantitative data. Especially for Fig. 5L, which pattern is the majority one or the allocation of each pattern

2). Fig. 8A shows that there are certain overlaps between H3K9me3 and H3K9ac in some areas. Can authors check the ratio of H3K9ac/H3K9me3 in active and silenced (or some poised) CVGs?

3). Line 333, "o" should be "or".

Staff Comments:

Preparing Revision Guidelines

Please return the manuscript within 60 days; if you cannot complete the modification within this time period, please contact me. If you do not wish to modify the manuscript and prefer to submit it to another journal, please notify me of your decision immediately so that the manuscript may be formally withdrawn from consideration by Microbiology Spectrum.

RESPONSE TO REVIEWERS

Reviewer #1 (Comments for the Author):

In this study, Michel-Todó and colleagues compare transcriptomic analysis with heterochromatin distribution in order to assess correlation between gene expression and chromatin accessibility in freshly derived wildtype clones.

The analysis by ChIP-Seq represents the novelty compared to the previous study from Rovira-Graells, 2012. Two clones from that work are re-analyzed here. Additionally, three fresh clones from a different 3D7 clones are added in the present work.

It is unclear to me the choice of the authors of re-analyzing only two of the three clones from the previous study. Why for the analysis they include the 3D7-B parental line instead of clone W4-1, which was generated in that study along with clones 1.2B and 10G? The use of three clones from the previous study versus three clones from the present study would make all the analysis more robust (see for example Figure 3).

This is especially true since the author state that due to different experimental setups they are unable to compare clones from the two studies for the transcriptomic analysis.

If the clone was discarded for any specific reason, the authors should discuss that in the text.

It is anyway my opinion that for the work to be robust enough, the analysis should be done on at least three clones per group of subclones.

We agree with the Reviewer that analyzing a larger number of subclones would have been optimal. However, we would like to clarify that our analysis to identify the heterochromatin changes associated with changes in the expression of clonally variant genes (CVGs) is not focused on the comparison of 3D7-A vs 3D7-B subclones, but rather on pairwise comparisons between subclones (we **modified the text in page 10, lines 212-3** to avoid possible misinterpretation. We agree that in the previous version of the manuscript this sentence may be prone to confusion). Therefore, with 5 subclones analyzed by ChIP-seq and transcriptomics, there are in total 10 possible independent pairwise comparisons that identified a total of 121 genes that are in a differential transcriptional state in at least one of the pairwise comparisons. While analyzing additional subclones may have identified a larger number of differentially expressed variant genes, we consider that 121 is a large enough number of genes to identify the dominant patterns of heterochromatin transitions between the active and silenced states of CVGs. To clarify these important aspects of the design of our study, in the revised manuscript we describe our selection of the subclones analyzed in more detail (**see new text in page 8, lines 167-71**).

Regarding the choice of subclones, we agree that including W4-1 appears to be logical. However, one of our main criteria for the selection was to include subclones with a large number of known or expected differentially expressed CVGs and covering a broad range of sexual conversion rates. W4-1 has the same molecular defect in *gdv1* as 10G and 1.2B and therefore does not produce gametocyte. Furthermore, based on our previous analysis in Rovira-Graells et al. 2012, there are a relatively small number of genes with large

transcriptional differences between W4-1 and the other subclones beyond the genes that already have differences among the subclones analyzed (less than 10).

Figure 2B and lines 213-214: "ChIP-seq analysis of H3K9me3 revealed a similar global distribution of heterochromatin between the five subclones, although differences were apparent at specific loci". These findings are described too briefly. The authors need to expand the description on what are the loci shown in the figure and what could be the significance of those differences. At least, in the figure, the genes marked with different chromatin status should be more clearly labeled.

Following the recommendation of the Reviewer, we added shades in Fig. 2B to label the regions in which prominent differences in heterochromatin were observed between subclones (**see revised Fig. 2B**). We also modified the text (**page 10, lines 232-3, and figure 2B legend in page 43**) to indicate that the regions shown in Fig. 2B are just representative examples. Additionally, we replaced the snapshot in the middle by a different chromosome region, rather than showing the same region as in the bottom panel at different magnification. This figure simply provides an overview of the H3K9me3 patterns observed. A detailed description of the differential peaks, the genes affected and their relation with transcriptional changes, with multiple snapshots for specific genes, is provided in the text and figures that follow.

Figure 3: why is this analysis performed only on 1.2B vs 10G, and not on the other three new clones?

Please note that the analysis was also performed for other direct pairwise comparisons, and presented in Supplementary Fig. 3. To make this clear, we **modified the text in page 11, lines 243-4**. This is also indicated at the beginning of the Fig. 3 legend. We preferred to show this analysis in a small main figure with only one representative comparison, and the rest in a Supplementary Figure, rather than making a large figure in the main text with all the pairwise comparisons, because the information provided by the different pairwise comparisons is somehow redundant. Of note, the next figure (Fig. 4), which includes data from all subclones, further develops the analysis of the association between transcriptional state and heterochromatin distribution.

Paragraph line 380-391: I think the results here are overinterpreted. It is a stretch to make a claim about *var* genes expression at single parasites by the heterochromatin distribution of the population.

We agree that since we did not directly measure *var* expression at the single cell level, the interpretation can be toned down. Therefore, we modified the text to make clear that unambiguous demonstration of simultaneous expression of multiple *var* genes or PfEMP1 forms in individual parasites would require additional experiments (**page 18, lines 423-28**). However, it is important to mention that while it is true that absence of heterochromatin at the promoter alone does not imply that a *var* gene is expressed, our statements are based

on the combination of heterochromatin analysis and transcriptomic analysis. While transcriptomic analysis indicates that several *var* genes are expressed in some of the subclones, heterochromatin profiling by ChIP-seq indicates that activation of each of the expressed *var* genes did not occur only in small subpopulations of parasites, because heterochromatin depletion was almost complete. If several *var* loci showed >80% heterochromatin depletion in the same subclone, this implies that individual parasites have more than one *var* gene in an active chromatin state simultaneously. As mentioned by Reviewer 3, this is consistent with previous observations from others.

Figure 7: It would be interesting to include in the figure a larger part of the upstream region of *gdv1*, to appreciate that the heterochromatin formation is only in the promoter of *gdv-as* and not in the one of *gdv1*.

We **modified Fig. 7** following this suggestion. A larger region is now shown for both *pfap2-g* and *gdv1*.

Line 445-446: it is again unclear to the reader why only those three clones were analyzed for H3K9ac and not all five available clones.

We **added a sentence in page 20, lines 483-4** to explain that this was an exploratory analysis, which justifies why we did not analyze this mark in all the subclones. In brief, the main focus of our work was on the distribution of heterochromatin in the active and silenced states of CVGs. We performed an exploratory analysis of the global distribution of H3K9ac in a limited number of subclones to determine if this analysis was able to identify a fully specific epigenomic signature of active and silenced CVGs. Since this was not the case and H3K9ac levels did not reveal a specific signature for active and silenced CVGs, we considered unnecessary to perform this costly analysis for the remaining subclones.

Reviewer #3 (Comments for the Author):

The submission from Lucas Michel-Todó et al characterized patterns of heterochromatin transitions of CVGs by checking the association between gene expression and heterochromatin mark, H3K9me3 and active mark H3K9ac in different clones of the human malaria parasite *Plasmodium falciparum*. The topic of CVG transition or switch in these parasites has been studied for many years and is critical for understanding this epigenetic mechanism. The results support that expression switches are probably caused by the expansion or retraction of heterochromatin domains. The authors also confirm the presence of multiple active *var* genes in some clones. Heterochromatin levels in the *gdv-1* ncRNA were positively correlated with the sexual conversion rates. Finally, the H3K9ac landscape was mostly not overlapped with heterochromatin domains. While the results are most convincing and add to the body of work on this topic, some (multiple *var* gene expression, H3K9ac landscape) are to a large extent confirmatory of previously published work that used similar approaches.

We agree with the Reviewer that some of the results are confirmatory of previously published work, and this is the reason why we focus mainly on the novel aspects of our results, i.e., the comparison of heterochromatin between the active and silenced state of CVGs at a global level. We added an additional reference reporting non-strict mutual exclusion of *var* genes expression (**new ref. 75, cited in page 18, line 428 and page 26, lines 625-6**).

1). While the heterochromatin patterns in Fig. 5 are mostly descriptive without quantitative data. Especially for Fig. 5L, which pattern is the majority one or the allocation of each pattern

We followed the recommendation of the Reviewer and in the revised manuscript we provide a quantification of the number of differentially expressed genes that follow each of the patterns described in the schematic in Fig. 5L. We added the number of genes following each type of transition in the text, together with an extended description (**new text in pages 14-5, lines 326-42**) and provide the list of genes under each pattern in a **new tab in Supplementary Table 5**. We also **improved the description of the patterns in the legend of Fig. 5L**. Please note that the other panels in Fig. 5 provide representative examples of the clusters in Fig. 4 (panels A-I), for which quantification of the number of genes in each cluster was already provided in the text, or refer to specific genes for which the active and silenced states had been compared before using ChIP-qPCR (panels J-K).

2). Fig. 8A shows that there are certain overlaps between H3K9me3 and H3K9ac in some areas. Can authors check the ratio of H3K9ac/H3K9me3 in active and silenced (or some poised) CVGs?

While in the full chromosome view (left panel of Fig. 8A) it may appear as if in some regions there is overlap between H3K9me3 and H3K9ac, the zoomed-in view at the right panel reveals that there is essentially no overlap between the two marks. The apparent small overlap disappears almost completely at higher magnification, and is likely attributable to the technical resolution of ChIP-seq or heterogeneity in the state of some CVGs (in spite of recent subcloning). To make this clearer, we added a new panel to the figure showing the distribution of H3K9ac and H3K9me3 at higher magnification. This new panel shows the association of the marks with the active or silenced states of individual CVGs (**new Fig. 8B and new text in page 20, line 490**). We also followed the Reviewer recommendation and analyzed the ratio of H3K9ac/H3K9me3 in active and silenced CVGs, and also in non-CVGs. The results of this new analysis, **added to the modified Fig. 8C** and mentioned in the text (**page 21, lines 503-4**), showed a very similar trend to the analysis of H3K9ac levels.

3). Line 333, "o" should be "or".

This has been corrected.

November 22, 2022

Prof. Alfred Cortés
Instituto de Salud Global Barcelona
Barcelona
Spain

Re: Spectrum03049-22R1 (Patterns of heterochromatin transitions linked to changes in the expression of *Plasmodium falciparum* clonally variant genes)

Dear Prof. Alfred Cortés:

Your manuscript has been accepted, and I am forwarding it to the ASM Journals Department for publication. You will be notified when your proofs are ready to be viewed.

Sincerely,

Laura Kirkman
Editor, Microbiology Spectrum

Journals Department
Supplemental Dataset S1: Accept
Supplemental Dataset S7: Accept
Supplemental Dataset S2: Accept
Supplemental Dataset S6: Accept
Supplemental Dataset S4: Accept
Supplemental Dataset S5: Accept
Supplemental Figures: Accept
Supplemental Dataset S3: Accept